# TRACTABLE DENDRITIC RNNS FOR IDENTIFYING UNKNOWN NONLINEAR DYNAMICAL SYSTEMS

## ABSTRACT

In many scientific disciplines, we are interested in inferring the nonlinear dynamical system underlying a set of observed time series, a challenging task in the face of chaotic behavior and noise. Previous deep learning approaches toward this goal often suffered from a lack of interpretability and tractability. In particular, the high-dimensional latent spaces often required for a faithful embedding, even when the underlying dynamics lives on a lower-dimensional manifold, can hamper theoretical analysis. Motivated by the emerging principles of dendritic computation, we augment a dynamically interpretable and mathematically tractable piecewise-linear (PL) recurrent neural network (RNN) by a linear spline basis expansion. We show that this approach retains all the theoretically appealing properties of the simple PLRNN, yet boosts its capacity for approximating arbitrary nonlinear dynamical systems in comparatively low dimensions. We introduce two frameworks for training the system, one based on fast and scalable variational inference, and another combining BPTT with teacher forcing. We show that the dendritically expanded PLRNN achieves better reconstructions with fewer parameters and dimensions on various dynamical systems benchmarks and compares favorably to other methods, while retaining a tractable and interpretable structure.

## 1 INTRODUCTION

For many complex systems in physics, biology, or the social sciences, we do not know or have only rudimentary knowledge about the dynamical system (DS) that may underlie those quantities that we can empirically observe or measure. Data-driven approaches aimed at automatically inferring the generating DS from time-series observations could therefore strongly support the scientific process, and various such methods have been proposed in recent years (Raissi et al., 2018; Zhu et al., 2021; Yin et al., 2021; Norcliffe et al., 2021; Mohajerin & Waslander, 2018; Karl et al., 2017; Chen et al., 2018; Strauss, 2020). However, due to the often high-dimensional, complex, chaotic, and inherently noisy nature of real-world DS, like the brain, weather-, or ecosystems, this remains a formidable challenge. Moreover, although the true DS may evolve on a lower-dimensional manifold in its state space, the system used for approximation usually needs to be of higher dimensionality to achieve a proper embedding (Takens, 1981; Sauer et al., 1991; Kantz & Schreiber, 2004). This is especially true when the approximating system is of a different functional form than the one that would most naturally describe the data generation process (but is unknown), for instance, when we attempt to approximate a system of exponential or trigonometric functions by polynomials.

In this work we sought to improve the capacity and expressiveness of a specific class of recurrent neural networks (RNNs), achieving agreeable solutions with fewer dimensions and parameters while retaining a set of desirable theoretical properties. Specifically, we build on piecewise-linear RNNs (PLRNNs) based on ReLU activation functions, for which fixed points, periodic orbits, and other dynamical properties can be derived analytically (Schmidt et al., 2021; Koppe et al., 2019), and for which dynamically equivalent continuous-time (ordinary differential equation, ODE) systems can be constructed (Monfared & Durstewitz, 2020b). Inspired by principles of dendritic computation in biological neurons (Fig. 1), each PLRNN unit was endowed with a set of nonlinear pre-processing subunits ("dendritic branches"), such that it effectively takes on the role of an equivalent much larger network. Mathematically, this comes down, in our case, to enhancing each latent unit with a linear spline basis expansion as popular in statistics (Hastie et al., 2009). Through this trick, we achieve

a powerful RNN which provides reconstructions of underlying nonlinear DS in lower-dimensional latent spaces than were needed by conventional PLRNNs. At the same time, model inference can be performed within the scalable framework of sequential variational auto-encoders (SVAE) (Archer et al., 2015; Girin et al., 2020; Krishnan et al., 2017), or with classical Back-Propagation-Through-Time (BPTT; Rumelhart et al. (1986)) augmented by teacher-forcing (TF; Williams & Zipser (1989); Pearlmutter (1990)). We further prove that these modifications preserve the mathematical and dynamical accessibility of the resulting system, e.g., such that fixed points, cycles, and their stability, can still be computed analytically.

Besides its effectiveness in capturing complex dynamical systems in fewer dimensions within a tractable framework, our approach highlights more generally how principles of dendritic signal processing may be harvested in the design of RNNs. Strongly nonlinear local computations are known for decades to occur within dendritic trees of biological neurons (Mel, 1994; Poirazi et al., 2003), but have hardly been exploited so far for machine learning models.

## 2  RELATED WORK

One class of DS reconstruction models attempts to discover governing equations from the flow field estimated from data through differencing the time series. Sparse Identification of Nonlinear Dynamics (SINDy), for instance, does so by sparsely regressing on a rich library of basis functions using the least absolute shrinkage and selection operator (LASSO) (Brunton et al., 2016; Rudy et al., 2017; de Silva et al., 2020). Other methods approximate the flow field using graph reconstruction via differential equations (Chen et al., 2017), sparse autoencoders (Heim et al., 2019), shallow multi-layer perceptrons reformulated as RNNs (Trischler & D'Eleuterio, 2016), or deep neural networks (Chen et al., 2018). Some works aimed at directly learning the system's underlying Hamiltonian (Chen et al., 2020; Greydanus et al., 2019). Generally, numerical derivatives obtained from time series tend to be more noise-prone than the time series observations themselves (Baydin et al., 2018; Chen et al., 2017; Raissi, 2018). This can be a problem particularly if only comparatively short trajectories were empirically observed or when the underlying systems are very high-dimensional, as in these cases the system's flow field may be (severely) under-sampled. Methods directly based on numerical derivatives also need to be augmented by other techniques, like delay embeddings (Kantz & Schreiber, 2004) or deep auto-encoders (Champion et al., 2019), if not all the system's dimensions were observed.

Various RNN architectures such as Long-Short-Term-Memory networks (LSTMs) (Zheng et al., 2017), Reservoir Computing (RC) (Pathak et al., 2018), or PLRNNs (Koppe et al., 2019; Schmidt et al., 2021) have been employed to infer DS directly from the observed time series without going through numerical derivatives. More recently, transformers (Shalova & Oseledets, 2020a;b) were used as black box approaches for DS prediction. Except for PLRNNs, however, all these systems, although optimized for DS reconstruction and prediction, rest on relatively complex model formulations that are not easy to tackle and analyze from a DS perspective (Fraccaro et al.). The ability to gain deeper insights into the specific DS properties and mechanisms of the recovered system is, however, often crucial for its applicability to science and engineering problems. Transformers, unlike RNNs, do not even constitute DS themselves (as they explicitly forgo any temporal recursions), and therefore are not directly amenable to DS theory tools. Moreover, most of these models, RC in particular, need very high-dimensional latent spaces, which further adds to their black-box nature.

Better interpretability and tractability is achieved by using PLRNNs (Koppe et al., 2019; Schmidt et al., 2021) or by (locally) linearizing nonlinear systems through ideas from Koopman operator theory (Azencot et al., 2020; Brunton et al., 2017; Yeung et al., 2017). In such systems, certain DS properties can be analytically accessed (Schmidt et al., 2021; Monfared & Durstewitz, 2020a), or the resulting equations can be more easily interpreted by a human reader (Heim et al., 2019). On the downside, usually one needs to move to very high dimensions to represent the DS in question properly. Here we aim to overcome this limitation by augmenting PLRNNs with linear basis expansions without altering their analytical accessibility.

Finally, probabilistic (generative) latent variable models such as state space models have been applied to the problem of posterior inference of latent state paths $z_t \sim p(z_t|x_{1:T})$ of DS given time series observations $\{x_{1:T}\}$ (Pandarinath et al., 2018; Ghahramani & Roweis, 1998; Durstewitz, 2017; Krishnan et al., 2017). The advantage here is that they also account for uncertainty in the model formulation or latent process itself and yield the full distribution over latent space variables (Karl

et al., 2017). For DS reconstruction, however, we need to move beyond posterior inference: We require that samples drawn from the model's prior distribution $p(\boldsymbol{z})$ after training exhibit the same temporal and geometric structure as those produced by the unknown DS.

Here we embed PLRNNs augmented with a linear spline expansion into a fully probabilistic, variational approach that scales well with system size by employing stochastic gradient variational Bayes (SGVB; (Kingma & Welling, 2014; Rezende et al., 2014)), thereby combining the advantages of the two classes of models reviewed above. On the other hand, we show that the model can also be efficiently trained by BPTT using a specific form of TF (Appx. 6.1).

## 3 MODEL FORMULATION AND THEORETICAL CONSIDERATIONS

### 3.1 PIECEWISE LINEAR RECURRENT NEURAL NETWORK (PLRNN)

Our approach builds on PLRNNs (Durstewitz, 2017; Koppe et al., 2019) because of their mathematical tractability (see Sec. 3.3). PLRNNs are defined by the $M$-dimensional latent process equation

$$\boldsymbol{z}_t = \boldsymbol{A}\boldsymbol{z}_{t-1} + \boldsymbol{W}\phi(\boldsymbol{z}_{t-1}) + \boldsymbol{h} + \boldsymbol{C}\boldsymbol{s}_t + \boldsymbol{\epsilon}_t, \tag{1}$$

which describes the temporal evolution of $M$-dimensional latent state vector $\boldsymbol{z}_t = (z_{1t} \dots z_{Mt})^T$. The self-connections of the units are represented by *diagonal* matrix $\boldsymbol{A} \in \mathbb{R}^{M \times M}$, whereas the connections between units are collected in *off-diagonal* matrix $\boldsymbol{W} \in \mathbb{R}^{M \times M}$, with the nonlinear activation function $\phi$ given by the rectified linear unit (ReLU) applied element-wise:

$$\phi(\boldsymbol{z}_{t-1}) = \max(0, \boldsymbol{z}_{t-1}). \tag{2}$$

Additionally, the PLRNN comprises a bias term $\boldsymbol{h} \in \mathbb{R}^M$, potential external inputs $\boldsymbol{s}_t \in \mathbb{R}^K$ weighted by $\boldsymbol{C} \in \mathbb{R}^{M \times K}$, and a Gaussian noise term $\boldsymbol{\epsilon}_t \sim \mathcal{N}(\boldsymbol{0}, \boldsymbol{\Sigma})$ with diagonal covariance $\boldsymbol{\Sigma}$. The PLRNN can be interpreted as a discrete-time neural rate model (Durstewitz, 2017), where the entries of $\boldsymbol{A}$ stand for the individual neurons' time constants, $\boldsymbol{W}$ for the synaptic connection strengths between neurons, and $\phi(z)$ for a (ReLU-shaped) voltage-to-spike-rate transfer function. The probabilistic latent RNN Eq. 1 is linked to the $N$-dimensional observed time series $(\boldsymbol{x}_t)_{t=1\dots T}$, $\boldsymbol{x}_t \in \mathbb{R}^N$, drawn from an underlying noisy DS, by an observation function (decoder model) which, in the simplest case, may take the linear Gaussian form

$$\boldsymbol{x}_t = \boldsymbol{B}\boldsymbol{z}_t + \boldsymbol{\eta}_t, \tag{3}$$

where $\boldsymbol{B} \in \mathbb{R}^{N \times M}$ represents a factor loading matrix and $\boldsymbol{\eta}_t \sim \mathcal{N}(\boldsymbol{0}, \boldsymbol{\Gamma})$ is Gaussian observation noise with diagonal covariance $\boldsymbol{\Gamma} \in \mathbb{R}^{N \times N}$.

### 3.2 DENDRITIC COMPUTATION AND SPLINE BASIS EXPANSION

Dendrites have long been known to play an active and important part in neural computation (Mel, 1994; 1999; Koch, 2004). Active, fast voltage-gated ion channels endow dendrites with strongly nonlinear behavior, giving rise for instance to dendritic $Ca^{2+}$ spikes that boost synaptic inputs (Schiller et al., 2000; Häusser et al., 2000). It has been suggested previously that different dendritic branches may constitute rather independent computational sub-units whose outputs are combined at the soma, as in a 2-layer neural network (Poirazi et al., 2003; Mel, 1993; 1994), an idea that received strong empirical support especially in recent years (Poirazi & Papoutsi, 2020). Here we mimic this functional setup by modeling dendritic processing through a linear combination of ReLU-type threshold-nonlinearities (Fig. 1), replacing Eq. 2 by

$$\phi(\boldsymbol{z}_{t-1}) = \sum_{b=1}^{B} \alpha_b \max(0, \boldsymbol{z}_{t-1} - \boldsymbol{h}_b), \tag{4}$$

with "dendritic input/output" slopes $\alpha_b \in \mathbb{R}$ and "activation" thresholds $\boldsymbol{h}_b \in \mathbb{R}^M$. As in real dendrites, where both ion channels and morphological structure are subject to learning (Poirazi & Papoutsi, 2020; Stemmler & Koch, 1999), we treat these as trainable parameters. We note that Eq. 4 inserted into model Eq. 1 takes the form of a linear spline basis expansion as popular in statistics (Hastie et al., 2009) for approximating arbitrary functions (Wahba, 1990; Storace & De Feo, 2004) in regression settings. For instance, such concepts have been frequently employed within data-analytical

models in neuroscience (Frank et al.; Huang et al.; Qian et al.), but never within the context of DS reconstruction enabling lower-dimensional solutions in mathematically tractable models.

To emphasize the connection to dendritic computation we call the system Eqs. 1, 3, 4, the *dendPLRNN*.

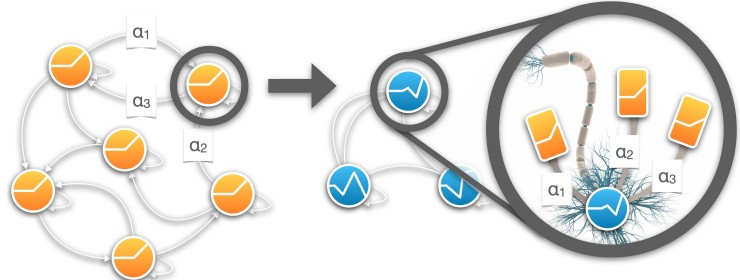

Figure 1: Inspired by principles of dendritic computation, our dendPLRNN extends each unit into a set of nonlinear branches connected to a soma, yielding single unit transfer functions with increased approximation capabilities. Image of dendrite from https://www.milad.no/blog/ (CC BY-SA 4.0).

### 3.3 MATHEMATICAL TRACTABILITY AND DYNAMICAL SYSTEMS INTERPRETATION

Sharp threshold-nonlinearities (like a ReLU) are a reasonable choice from a neurobiological perspective, as dendrites naturally give rise to this threshold-type behavior (Mel, 1999; Koch, 2004). Another important consideration in choosing this particular form, however, was that it preserves all the theoretically appealing properties of a PLRNN, as we will formally establish below: For PLRNNs fixed points and cycles can be explicitly computed (Schmidt et al., 2021; Koppe et al., 2019), and they can be translated into dynamically equivalent continuous-time systems (Monfared & Durstewitz, 2020b), properties which profoundly ease the analysis of trained systems from a DS perspective. This is crucial for application in the sciences, where we are specifically interested in understanding the underlying system's dynamics. For PLRNNs, precise connections between the long-term behavior of the system and that of its gradients have also been established (Schmidt et al., 2021). Finally, PLRNNs belong to the class of continuous piecewise-linear (PWL) maps, for which many important types of bifurcations have been well characterized (Feigin, 1995; Hogan et al., 2007; Patra, 2018) (cf. (Monfared & Durstewitz, 2020a) for an overview). Bifurcations are essential to understand how geometrical and topological properties of the system's state space depend on its parameters or could be controlled, and hence are also important to characterize or improve the training process itself (Doya, 1992; Pascanu et al., 2013; Saxe et al., 2014) or to understand properties of trained systems (Maheswaranathan et al., 2019b;a) .

Our first proposition, therefore, assures that by the particular form of basis expansion introduced in Eq. 4, the system will remain within the class of continuous PWL maps:

**Proposition 1.** *The model defined through Eq. 1 and Eq. 4 constitutes a continuous PWL map.*

The proof essentially straightforwardly follows from the model's definition as a linear spline basis expansion in each unit, but is formally provided in Appx. 6.5.4.

While Proposition 1 is all we need to ensure we can harvest all previously established results on PLRNNs in particular, and on continuous PWL maps more generally, it is revealing to note that any dendPLRNN (Eqs. 1, 4) can be rewritten as a conventional PLRNN, as stated in the following theorem:

**Theorem 1.** *Any $M$-dimensional dendPLRNN as defined in Eqs. 1, 4, can always be rewritten as a $M \times B$-dimensional "conventional" PLRNN of the form*

$$\hat{z}_t = \tilde{A}\hat{z}_{t-1} + \tilde{W} \max(0, \hat{z}_{t-1}) + \hat{h}_0 + \tilde{C}s_t + \tilde{\epsilon}_t. \tag{5}$$

*Proof.* Straightforward by construction, see Appx. 6.5.5. □

This theorem highlights why the dendPLRNN will allow to reduce the dimensionality of the reconstructed system, as it suggests we may often be able to reformulate a high-dimensional PLRNN in terms of an equally powerful lower-dimensional dendPLRNN. In Appx. 6.5.1 we also spell out the exact computation of fixed points and $k$-cycles for the dendPLRNN.

Finally, the unboundedness of the PLRNN's latent states due to the ReLU function can cause divergence problems in training. The dendPLRNN, on the other hand, offers a simple and natural way to contain the latent states without violating the basic model description above, as established in the following theorem:

**Theorem 2.** *For each basis $\{\alpha_b, \boldsymbol{h}_b\}$ in Eq. 4 of a dendPLRNN let us add another basis $\{\alpha_b^*, \boldsymbol{h}_b^*\}$ with $\alpha_b^* = -\alpha_b$ and $\boldsymbol{h}_b^* = \boldsymbol{0}$. Then, for $\sigma_{\max}(\boldsymbol{A}) < 1$, any orbit of this "clipped" dendPLRNN (Eq. 10) will remain bounded.*

*Proof.* See Appx. 6.5.6. □

Appx. 6.5 collects further theoretical results, assuring, for instance, that the manifold attractor regularization employed here (see next section) does not interfere with the results above (Proposition 2).

## 3.4 Training the dendPLRNN

To infer the parameters $\theta = \{\boldsymbol{A}, \boldsymbol{W}, \boldsymbol{h}, \boldsymbol{C}, \boldsymbol{\Sigma}, \boldsymbol{B}, \boldsymbol{\Gamma}, \{\boldsymbol{\alpha}_b, \boldsymbol{h_b}\}\}$ of the dendPLRNN (Eq. 1, 3, 4) from observed data, we apply two different training strategies: First, a fast and scalable variational inference (VI) algorithm which maximizes the Evidence Lower Bound (ELBO) $\mathcal{L}(\theta, \phi; \boldsymbol{x}) := \mathbb{E}_{q_\phi}[\log(p_\theta(\boldsymbol{x}|\boldsymbol{z})] - \mathrm{KL}[q_\phi(\boldsymbol{z}|\boldsymbol{x})||p_\theta(\boldsymbol{z})]$ using the reparameterization trick (Kingma & Welling, 2014), and convolutional neural networks (CNNs) for parameterizing the encoder model $q_\phi(\boldsymbol{z}|\boldsymbol{x})$ (see Appx. 6.1 for details). Furthermore, as proposed in Schmidt et al. (2021), to efficiently capture DS at multiple time scales, we add a regularization term to the ELBO that encourages the mapping of slow time constants and long-range dependencies (so-called *manifold attractor regularization*, see Eq. 6, with regularization factor $\lambda$). Second, we employ "classical" BPTT with a variant of teacher forcing (TF) (Williams & Zipser, 1989; Pearlmutter, 1990). TF here means that the first $N$ latent states $z_{k,l\tau+1}, k \leq N$, were replaced by observations $x_{k,l\tau+1}$ at times $l\tau + 1, l \in \mathbb{N}_0$, where $\tau \geq 1$ is the forcing interval (for details, see Appx. 6.1). All code used in here is made freely available at [placeholder].

## 4 Experiments

### 4.1 Performance measures

In DS reconstruction, we aim to capture *invariant* properties of the underlying DS like its geometrical and temporal structure. To evaluate the quality of the reconstructions w.r.t. *geometrical properties* we employed a Kullback-Leibler divergence ($D_{\mathrm{stsp}}$) that quantifies agreement in attractor geometries (more details in Appx. 6.2), as first suggested in Koppe et al. (2019) (see also Schmidt et al. (2021)). Specifically, this measure evaluates the overlap between the observed data distribution $p(\boldsymbol{x}^{\mathrm{obs}})$ and the distribution $p(\boldsymbol{x}^{\mathrm{gen}}|\boldsymbol{z}^{\mathrm{gen}})$ generated from model simulations (i.e., with $\boldsymbol{z}^{\mathrm{gen}} \sim p_\theta(\boldsymbol{z})$ after model training) across state *space* (not time!). Since this measure as originally defined in Koppe et al. (2019) is expensive to compute, for the high-dimensional benchmark DS we used another approximation, details of which are given in Appx. 6.2. $D_{\mathrm{stsp}}$ is evaluated on a set of 100 trajectories, pulled from the learned distribution over initial conditions, with 1000 time steps each. To assess the agreement in *temporal structure*, a dimension-wise, Gaussian-kernel-smoothed power spectrum correlation (PSC) between ground truth and model-generated trajectories was used (see Appx. 6.2). Finally, we also computed a 20-step-ahead prediction error along test set trajectories (see Appx. 6.2), although not of primary interest in the context of DS reconstruction.

### 4.2 DS benchmarks used for evaluation

We evaluated our approach and the specific role of the basis expansion on five different types of challenging DS benchmarks.

First, the famous 3d chaotic Lorenz attractor (Lorenz-63) originally proposed by Lorenz (1963) (formally defined in Appx. 6.4) has become a popular benchmark for DS reconstruction algorithms. Fig. 2a (l.h.s.) illustrates true (blue) and reconstructed (orange) time series from this system, while the r.h.s. illustrates the chaotic attractor's geometry in its 3d state space for both the ground truth (blue) and reconstructed (orange) systems. It is important to note that both the time and state space graphs are not merely ahead predictions from the dendPLRNN but are produced by *simulating* the trained dendPLRNN from some initial condition. This illustrates that the dendPLRNN has captured the temporal and geometrical structure of the original Lorenz-63 system in its own governing equations. Moreover, computing analytically (see Appx. 6.5.1) the fixed points of the reconstructed system, we see that their positions in state space agree well with those of the true system.

Second, a 3d biophysical model of a bursting neuron (see Eq. 15 in Appx. 6.4; Durstewitz (2009)) highlights another aspect of DS reconstruction: Besides an equation for membrane voltage ($V$), the model consists of one very fast ($n$) and one slow ($h$) variable that control the gating of the model's ionic conductances. This produces fast spikes that ride on top of a much slower oscillation, making this system challenging to reconstruct. One such successful dendPLRNN reconstruction is illustrated in Fig. 2b (orange) together with time graphs and state space representations of the true system (blue).

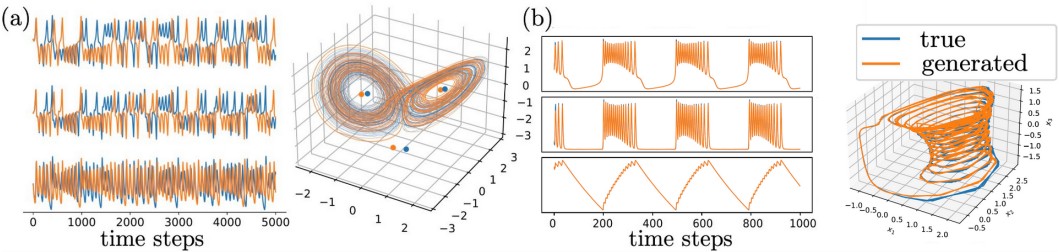

Figure 2: Examples of low-dimensional model reconstructions: (a) Time series (left) and state space trajectories (right) for the original Lorenz-63 chaotic attractor and simulations produced by a dendPLRNN trained with VI ($B = 20$, $M = 15$, $\lambda = 1$, $M_{\mathrm{reg}}/M = 0.5$). Dots indicate true and reconstructed fixed points. (b) Same for the bursting neuron model, produced by a dendPLRNN trained with TF ($B = 47$, $M = 26$, $\tau = 5$). Note that the bursting is a complex limit cycle but *non-chaotic*.

Third, the Lorenz-96 weather model is an example of a higher-dimensional, spatially organized chaotic system with local neighborhood interactions that can be extended to arbitrary dimensionality (Eq. 18 in Appx. 6.4). It has also been used more widely for benchmarking DS reconstruction algorithms. For our experiments we employed a 10-dimensional spatial layout. Fig. 3a illustrates time graphs for selected dimensions (l.h.s.), the full evolving spatio-temporal pattern (center), and examples of power spectra (r.h.s.) for both the ground truth system (blue) and an example reconstruction (orange). The spatio-temporal characteristics of the true and the dendPLRNN-generated time series tightly agree.

Fourth, as another high-dimensional example we used a neural population model with structured connectivity tuned to produce coherent chaos (Landau & Sompolinsky, 2018), from which we produced 50d observations (see Appx. 6.4 for details). Fig. 3b provides example time series (l.h.s.), full spatio-temporal patterns (center), and overlaid power spectra (r.h.s.) for time series drawn from the true system (blue) and those simulated by a trained dendPLRNN (orange). Again there is a tight agreement, and again we emphasize that - like in all the other examples - these are not mere model ahead-predictions but fully simulated from some random initial condition.

Finally, we studied a real-world dataset consisting of electroencephalogram (EEG) recordings from human subjects, described in more detail with results (Fig.***) in Appx. ***.

### 4.3 BASIS EXPANSION IMPROVES DS RECONSTRUCTION AND LOWERS PARAMETER COSTS

While, in theory, the dendPLRNN is equivalent to a larger PLRNN without basis expansion, in practice the smaller basis-expanded models trained more successfully. Fig. 4a summarizes our observations for the VI algorithm on the impact of the basis expansion using the Lorenz-63 DS as an example (see Fig. S3 for further examples): Both the 20-step ahead prediction error as well

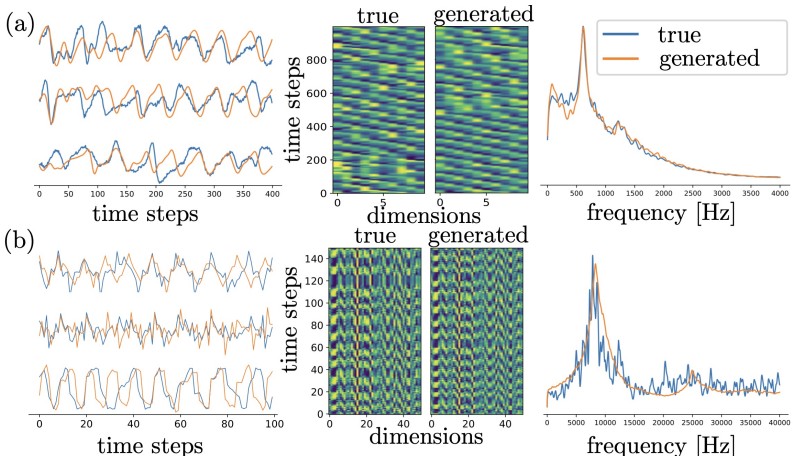

Figure 3: Examples of high-dimensional model reconstructions: (a) Time series (left), spatio-temporal evolution (center), and power spectra (right) for the true 10d Lorenz-96 system and for dendPLRNN simulations ($B = 50$, $M = 30$, $\lambda = 1.0$, $M_{\mathrm{reg}}/M = 1.0$). (b) Same for a 50d neural population model producing coherent chaos ($B = 5$, $M = 12$, $\lambda = 1.0$, $M_{\mathrm{reg}}/M = 0.2$).

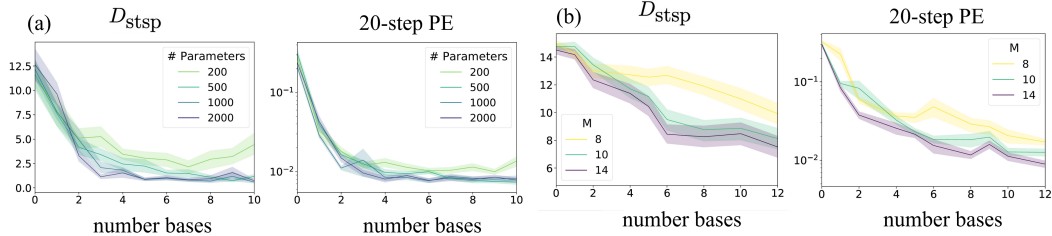

Figure 4: Effect of basis expansion for dendPLRNN trained by VI. (a) Agreement in attractor geometries (left) and 20-step ahead prediction error (right) for the Lorenz-63 system as a function of the number of bases ($B$) for fixed numbers of total parameters. (b) Agreement in attractor geometries (left) and 20-step ahead prediction error (right) for the Lorenz-63 system as a function of the number of bases ($B$) for different numbers of latent states ($M$).

as the geometrical reconstruction quality as assessed by $D_{\mathrm{stsp}}$ profoundly improve with the size $B$ of the basis expansion *even for the same total number of trainable model parameters* (given by $M(M + 1 + B + N) + B$, where $N$ is the dimensionality of the observed data). Hence, as conjectured in Sec. 3, the basis expansion yields better reconstructions at no additional costs in terms of numbers of model parameters. Fig. 4b looks at the impact of the basis expansion from the angle of dimensionality reduction by systematically varying the number of bases $B$ and latent states $M$ for the Lorenz-63 DS: Following the curves horizontally, it becomes clear that the basis expansion enables to reduce the model's overall dimensionality without compromising performance.

## 4.4 MODEL COMPARISONS

We compared our model to the PLRNN without the dendritic expansion and three other algorithms purpose-tailored for DS reconstruction: First, SINDy (Brunton et al., 2016) aims to reconstruct the governing equations by approximating numerical derivatives (obtained by differencing the time series, and applying a variance regularization to reduce noise) through a large library of polynomial basis functions. Sparse (LASSO) regression is used to pick out the right terms from the library (we used the PySINDy implementation (de Silva et al., 2020) with multinomials up to sixth order). Second, Vlachas et al. (2018) used a hybrid of truncated LSTMs and mean-field stochastic models based on Ornstein-Uhlenbeck processes (LSTM-MSM) to approximate the true system's flow estimated from observed time series. Third, Pathak et al. (2018) built on reservoir computing (RC) for their approach with reservoir parameters chosen to satisfy the "echo state property" (Jaeger & Haas, 2004). For

Table 1: Comparison of dendPLRNN (Ours) trained by VI or BPTT+TF, RC (Pathak et al., 2018), LSTM-MSM (Vlachas et al., 2018), and SINDy (Brunton et al., 2016) on 4 DS benchmarks and one experimental dataset (top) and 3 challenging data situations (bottom). Values are mean $\pm$ SEM.

| Dataset / Setting | Method | PSC | $D_{stsp}$ | 20-step PE | Dynamical variables | Parameters |
|---|---|---|---|---|---|---|
| Lorenz | dendPLRNN VI | $0.997 \pm 0.001$ | $0.80 \pm 0.25$ | $2.1e{-}3 \pm 0.2e{-}3$ | 22 | 1032 |
| | dendPLRNN TF | $0.997 \pm 0.002$ | $0.13 \pm 0.18$ | $9.2e{-}5 \pm 2.8e{-}5$ | 22 | 1032 |
| | RC | $0.991 \pm 0.001$ | $0.24 \pm 0.05$ | $1.2e{-}2 \pm 0.1e{-}3$ | 345 | 1053 |
| | LSTM-MSM | $0.985 \pm 0.004$ | $0.85 \pm 0.07$ | $1.2e{-}2 \pm 0.1e{-}3$ | 29 | 1035 |
| | SINDy | $\mathbf{0.998 \pm 0.0003}$ | $\mathbf{0.04 \pm 0.01}$ | $\mathbf{6.8e{-}5 \pm 0.2e{-}5}$ | 3 | 252 |
| Lorenz-96 | dendPLRNN VI | $0.987 \pm 0.001$ | $0.10 \pm 0.01$ | $3.1e{-}1 \pm 0.9e{-}1$ | 42 | 4384 |
| | dendPLRNN TF | $\mathbf{0.998 \pm 0.0001}$ | $\mathbf{0.04 \pm 0.01}$ | $4.1e{-}2 \pm 0.8e{-}2$ | 50 | 4480 |
| | RC | $0.986 \pm 0.008$ | $0.25 \pm 0.17$ | $7.1e{-}1 \pm 0.1e{-}2$ | 440 | 4400 |
| | LSTM-MSM | $0.993 \pm 0.002$ | $0.23 \pm 0.03$ | $8.2e{-}1 \pm 0.3e{-}2$ | 62 | 4384 |
| | SINDy | $0.996 \pm 0.001$ | $0.06 \pm 0.003$ | $6.3e{-}2 \pm 0.1e{-}3$ | 10 | 27410 |
| Bursting Neuron | dendPLRNN VI | $0.55 \pm 0.03$ | $7.5 \pm 0.4$ | $6.1e{-}1 \pm 0.1e{-}1$ | 26 | 2052 |
| | dendPLRNN TF | $\mathbf{0.76 \pm 0.04}$ | $2.9 \pm 1.3$ | $6.1e{-}2 \pm 2.2e{-}2$ | 26 | 2040 |
| | RC | $0.51 \pm 0.01$ | $5.1 \pm 0.6$ | $8.6e{-}2 \pm 0.1e{-}2$ | 711 | 2133 |
| | LSTM-MSM | $0.54 \pm 0.02$ | $\mathbf{2.83 \pm 0.36}$ | $\mathbf{3.9e{-}2 \pm 0.1e{-}2}$ | 45 | 2166 |
| | SINDy | diverging | diverging | diverging | 3 | 252 |
| Neural Population Model | dendPLRNN VI | $0.45 \pm 0.05$ | $0.56 \pm 0.05$ | $\mathbf{0.82 \pm 0.09}$ | 12 | 821 |
| | dendPLRNN TF | $\mathbf{0.51 \pm 0.01}$ | $\mathbf{0.19 \pm 0.02}$ | $1.53 \pm 0.03$ | 75 | 9990 |
| | RC | $0.30 \pm 0.05$ | $0.95 \pm 0.19$ | $1.82 \pm 0.82$ | 50 | 2500 |
| | LSTM-MSM | $0.45 \pm 0.03$ | $0.43 \pm 0.02$ | $1.02 \pm 0.02$ | 56 | 848 |
| | SINDy | diverging | diverging | diverging | 50 | 66300 |
| EEG | dendPLRNN VI | $0.80 \pm 0.01$ | $27.9 \pm 3.6$ | $0.56 \pm 0.046$ | 117 | 27194 |
| | dendPLRNN TF | $\mathbf{0.936 \pm 0.017}$ | $\mathbf{4.7 \pm 2.7}$ | $\mathbf{0.267 \pm 0.013}$ | 128 | 27058 |
| | RC | $0.81 \pm 0.01$ | $21.2 \pm 2.2$ | $5.4 \pm 0.2$ | 448 | 28672 |
| | LSTM-MSM | $0.84 \pm 0.005$ | $19.9 \pm \mathbf{1.8}$ | $2.0 \pm 0.5$ | 168 | 27728 |
| | SINDy | diverging | diverging | diverging | 64 | 133120 |
| Low amount of data | dendPLRNN VI | $0.967 \pm 0.007$ | $4.36 \pm 0.10$ | $2.8e{-}2 \pm 0.2e{-}2$ | 22 | 1032 |
| | dendPLRNN TF | $0.97 \pm 0.04$ | $6.9 \pm 5.3$ | $1.5e{-}2 \pm 0.9e{-}2$ | 22 | 1032 |
| | RC | $0.68 \pm 0.05$ | $5.74 \pm 0.11$ | $4.1e{+}5 \pm 1.2e{+}5$ | 345 | 1053 |
| | LSTM-MSM | $0.960 \pm 0.006$ | $6.06 \pm 0.37$ | $2.1e{-}1 \pm 0.3e{-}2$ | 29 | 1035 |
| | SINDy | $\mathbf{0.998 \pm 0.0003}$ | $\mathbf{0.04 \pm 0.01}$ | $\mathbf{6.8e{-}5 \pm 0.2e{-}5}$ | 3 | 252 |
| Partially observed | dendPLRNN VI | $0.940 \pm 0.006$ | $12.6 \pm 1.0$ | $6.5e{-}2 \pm 1.4e{-}2$ | 22 | 1032 |
| | dendPLRNN TF | $\mathbf{0.993 \pm 0.003}$ | $\mathbf{0.54 \pm 0.16}$ | $\mathbf{5.3e{-}3 \pm 0.2e{-}3}$ | 22 | 1032 |
| | RC | $0.981 \pm 0.001$ | $2.92 \pm 0.08$ | $7.6e{-}3 \pm 0.1e{-}3$ | 345 | 1053 |
| | LSTM-MSM | $0.934 \pm 0.005$ | $6.06 \pm 0.37$ | $2.3e{-}2 \pm 0.3e{-}2$ | 29 | 1035 |
| | SINDy | $0.974 \pm 6e-4$ | $17.5 \pm 0.4$ | $5.1e{-}2 \pm 0.4e{-}2$ | 3 | 252 |
| High noise | dendPLRNN VI | $0.973 \pm 0.006$ | $4.9 \pm 0.75$ | $3.5e{-}2 \pm 0.1e{-}2$ | 22 | 1032 |
| | dendPLRNN TF | $\mathbf{0.995 \pm 0.002}$ | $\mathbf{0.4 \pm 0.13}$ | $4.6e{-}2 \pm 0.4e{-}3$ | 22 | 1032 |
| | RC | $0.988 \pm 0.001$ | $2.33 \pm 0.21$ | $3.1e{-}2 \pm 0.2e{-}2$ | 345 | 1053 |
| | LSTM-MSM | $0.967 \pm 0.006$ | $1.19 \pm 0.27$ | $3.3e{-}2 \pm 0.2e{-}2$ | 29 | 1035 |
| | SINDy | $0.984 \pm 0.005$ | $1.01 \pm 0.05$ | $\mathbf{2.3e{-}3 \pm 0.1e{-}4}$ | 3 | 252 |

higher-dimensional systems, a spatially arranged set of reservoirs with local neighborhood relations is employed. For all these systems, optimized hyper-parameters were used as reported by the authors. For our own system, the dendPLRNN, we also performed a grid search for optimal hyper-parameters $\lambda_{reg}$, $\tau_{TF}$, $M$, and $B$ (see Appx. 6.1 and Table S1 for details). For all four methods, to the degree possible we tried to ensure roughly the same number of trainable parameters (see Table S2).

Results for all four models on all five DS benchmarks employed here are summarized in the upper part of Table S2, using the temporal and geometrical reconstruction measures introduced in Sec. 4.2 (as well as a 20-step-ahead prediction error for comparison). To produce this table, 100,000 time steps for both training and testing were simulated from each ground truth system, all dimensions were standardized to have zero mean and unit variance, and process noise and observation noise (with 1% of the data variance) were added while simulating the (now stochastic) differential equations, and after drawing the observations, respectively (see Appx. 6.4 for further methodological details). To produce statistics, each method was run from a total of 20 randomly chosen initial conditions for the parameters. We also tested all four methods on the real EEG data and on challenging data situations produced using the Lorenz-63 system (Fig. 2a), with either short time series of just 1000 time steps, only partial observations (just state variable $x$ in Eq.14 in Appx. 6.4), or high process and high observation noise (drawing from a Gaussian with $d\epsilon \sim \mathcal{N}(0, 0.1dt \times \mathbf{I})$ for the process noise as described in Appx. 6.4, and using 10% of the observation variance, respectively). SINDy cannot naturally handle missing observations, as it has no latent variables but formulates the model directly in terms of the observations. Therefore, for the partially observed system, we used delay embedding (Takens, 1981; Sauer et al., 1991) to create a 3d dataset, adding two time-lagged versions of $x$ as coordinates.[1]

A general observation is that indeed all four models are quite powerful for reconstructing the underlying DS. However, in most comparisons the dendPLRNN had an edge over the other methods,

---

[1]We point out that this may already impose a restriction for methods like SINDy as one moves to very high-dimensional systems.

or came out second after SINDy, especially when trained by BPTT+TF. SINDy tends to outperform the dendPLRNN on the Lorenz-63 DS, but it completely fails on the bursting-neuron and population model examples, and on the real EEG data, and generally becomes comparatively slow to train on high-dimensional systems. This can be explained by the fact that SINDy already has the correct functional form for the Lorenz-63 (and also Lorenz-96) DS: Both of these have a strictly polynomial form (see Eq. 14 and Eq. 18 in Appx. 6.4), and SINDy works with a set of polynomial library functions to begin with. Hence, SINDy only needs to pick out the right terms from its expansion to succeed, giving it a clear advantage on these model systems by design. On the other hand, as revealed in Table S2, it completely fails on systems which have a different (non-polynomial) functional form (or when the true form, as in the EEG case, is simply not known). SINDy therefore appears less suitable as a general framework for DS reconstruction if an appropriate library of basis functions cannot be specified a priori, unlike the other methods.

While our conclusion is that essentially all of the three tested models LSTM-MSM, RC, and dend-PLRNN are suitable for reconstruction of arbitrary DS even in very challenging data situations (Table S2, bottom), LSTM-MSM and RC performed worse on average and have other profound disadvantages compared to our method: First, they are quite complex in their architectures and hence not easily interpretable, i.e. much harder to track and analyze mathematically.[2] In contrast, as summarized in Sec. 3.3, the dendPLRNN is a continuous PWL map and as such comes with a huge bulk of already existing theoretical results (Schmidt et al., 2021; Monfared & Durstewitz, 2020b;a), as well as with mathematical tractability (see Fig. 2a and Appx. 6.5.1). On top, the dendPLRNN achieves reconstruction of all DS in (much) lower dimensions than the RC or LSTM-MSM (see Table S2), further adding to its better interpretability. Second, by embedding the dendPLRNN within a SVAE (Archer et al., 2015) framework we also obtain uncertainty estimates on the state trajectories and can perform posterior inference, features that the other models lack.

## 5  CONCLUSIONS

In this work we augmented PLRNNs (Durstewitz, 2017; Koppe et al., 2019) by a linear spline basis expansion inspired by principles of dendritic computation. We show mathematically that by doing so we remain within the theoretical framework of continuous PWL maps and hence can harvest a huge bulk of existing DS theory (Sec. 3.3), while at the same time achieving better performance with less parameters and in lower dimensions. Another contribution of this work is transferring the PLRNN into the framework of SVAEs which allow for fast and scalable inference and training. These are two key advantages from both a scientific perspective where mechanistic insight and understanding of the system under study are sought, and a prediction perspective where we are also interested in uncertainty estimates.

We close by pointing out two open issues: First, somewhat surprisingly, the BPTT+TF approach to model training clearly outperformed the more sophisticated VI approach. This could be rooted in suboptimal encoder models or in suboptimal sampling from the approximate posterior: While BPTT+TF assesses longer bits of trajectory during optimization, in VI single time-point samples are drawn and the temporal consistency is ensured only through the Kullback-Leibler term in the ELBO. Other more expressive yet still fast to compute encoder models, e.g., based on normalizing flows (Rezende & Mohamed, 2015), may boost performance. Smart initialization techniques (Talathi & Vartak, 2016) or specific annealing and curriculum training protocols (as used in Koppe et al. (2019)) are other amendments to consider. Second, we felt that quantitative measures for assessing the quality of DS reconstructions in high-dimensional, high-noise situations are an interesting research topic in their own right. It is known that "classical" DS measures like Lyapunov spectra or correlation dimensions (Kantz & Schreiber, 2004) are very hard to robustly assess for higher-dimensional or more noisy systems (Schreiber & Kantz, 1996), and are often not even known for comparatively simple models. Yet, the geometrical and temporal measures employed here come with their own pitfalls, some of them alluded to in Appx. 6.2.

ACKNOWLEDGMENTS

---

[2]This is especially true for RC. Moreover, the fact that only the weights of the linear output layer are trainable while the recurrent connections within the reservoirs are static, may raise the question of what precisely is learnt in terms of dynamics if the reservoirs themselves cannot adapt to the DS at hand.

**Ethics statement** The current work performs theoretical analysis and basic research on a novel type of RNN architecture, which is evaluated exclusively on a set of simulated physical and biological systems (no human or animal subjects involved, no privacy concerns etc.). Although it is conceivable that the current models, training algorithms, and results may be used to develop prediction algorithms in sensitive domains (like medical time series or tracking consumer data), there are no immediate ethical implications from this work as far as we can see.

**Reproducibility statement** All theoretical results in this paper were carefully and thoroughly proven, with all proofs and detailed derivations available in the Appendix. Likewise, we will make available all code used in the empirical section in a way that will allow others to easily reproduce the results from this paper. This means we will include everything, starting with the code for the benchmark models and simulations, the simulated time series data used for evaluation themselves, the code for our own model and training algorithms, up to the meta-files that produce the figures in this work, on our lab github site. All of this will be clearly documented.

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

# 6 APPENDIX

## 6.1 FURTHER METHODOLOGICAL DETAILS

**Manifold Attractor Regularization** As proposed in Schmidt et al. (2021), to encourage the discovery of long-term dependencies and slow time scales in the data, a subset of $M_{\mathrm{reg}} \leq M$ states was regularized by adding the following term to the ELBO for the VI approach:

$$\mathcal{L}_{\mathrm{reg}} = \lambda \left( \sum_{i=1}^{M_{\mathrm{reg}}} (A_{ii} - 1)^2 + \sum_{i=1}^{M_{\mathrm{reg}}} \sum_{j \neq i}^{M} (W_{ij})^2 + \sum_{i=1}^{M_{\mathrm{reg}}} h_i^2 \right). \tag{6}$$

This regularization pushes the regularized subset of states toward a continuous set of marginally stable fixed points that tends to form an attracting manifold in the full state space, which supports the learning of systems with widely differing time scales, such as the bursting neuron model (cf. Sec. 4). We found that for all datasets this regularization significantly helped to discover the underlying dynamics. To put it on equal grounds with the regularization term, the ELBO was divided by the number of time steps $T$ of a given batch. Regularization settings used are summarized in Table S1 along other hyper-parameter settings.

**BPTT-TF** To train a deterministic version of the dendPLRNN, we employ BPTT with a scheduled version of TF (Williams & Zipser, 1989; Pearlmutter, 1990). To do so, we choose an "identity-mapping" for the observation model $\hat{\boldsymbol{x}}_t = \mathcal{I}\boldsymbol{z}_t$, where $\mathcal{I} \in \mathbb{R}^{N \times M}$ with $\mathcal{I}_{kk} = 1$ if $k \leq N$ and zeroes everywhere else. This allows us to regularly replace latent states with observations to "recalibrate" the model and break trajectory divergence in case of chaotic dynamics. Consider a time series $\{\boldsymbol{x}_1, \boldsymbol{x}_2, \cdots, \boldsymbol{x}_T\}$ generated by a DS we want to reconstruct. At times $l\tau + 1$, $l \in \mathbb{N}_0$, where $\tau \geq 1$ is the forcing interval, we replace the first $N$ latent states by observations $\hat{z}_{k,l\tau+1} = x_{k,l\tau+1}$, $k \leq N$. The remaining latent states, $\hat{z}_{k,l\tau+1} = z_{k,l\tau+1}$, $k > N$, remain unaffected by the forcing. This means that we optimize the dendPLRNN such that a subspace of the latent space directly maps to the observed time series variables. The forcing interval $\tau$ is a hyperparameter, with optimal settings varying depending on the dataset. The settings we chose are summarized in Table S1. With $\mathcal{F} = \{l\tau + 1\}_{l \in \mathbb{N}_0}$, the dendPLRNN updates can then be written as

$$\boldsymbol{z}_{t+1} = \begin{cases} dendPLRNN(\tilde{\boldsymbol{z}}_t) & \text{if } t \in \mathcal{F} \\ dendPLRNN(\boldsymbol{z}_t) & \text{else} \end{cases}. \tag{7}$$

The loss is calculated prior to the forcing, such that $\mathcal{L}_t = \|\boldsymbol{x}_t - \mathcal{I}\boldsymbol{z}_t\|_2^2$ for every time step. To improve performance we employ a mean-centred dendPLRNN (for details see next paragraph). In the evaluation phase, the trained dendPLRNN is simulated freely without any forcing. As the model is deterministic, the initial condition $z_1 = [\boldsymbol{x}_1, \boldsymbol{L}\boldsymbol{x}_1]^\mathsf{T}$ is estimated from the first data point $\boldsymbol{x}_1$ with a matrix $\boldsymbol{L} \in \mathbb{R}^{(M-N) \times N}$ which is jointly learned with the other model parameters.

**Mean-Centered dendPLRNN** Layer normalization has recently been developed as a way of significantly improving RNN training (Ba et al., 2016). Here we adapt the idea of layer normalization to the piecewise-linear nature of our dendPLRNN. Instead of fully standardizing the latent states at every time step before applying the activation function, we only mean-center them:

$$\boldsymbol{z}_t = \boldsymbol{A}\boldsymbol{z}_{t-1} + \boldsymbol{W}\phi\big(\mathcal{M}(\boldsymbol{z}_{t-1})\big) + \boldsymbol{h}_0, \tag{8}$$

where $\phi(\cdot)$ is given in Eq. 4 and $\mathcal{M}(\boldsymbol{z}_{t-1}) = \boldsymbol{z}_{t-1} - \mu_{t-1} = \boldsymbol{z}_{t-1} - \mathbf{1}\frac{1}{M}\sum_{j=1}^{M} z_{j,t-1}$, where

$\mathbf{1} \in \mathbb{R}^M$ is a vector of ones. Note that this mean-centering is linear and can be rewritten as a matrix-multiplication

$$\mathcal{M}(\boldsymbol{z}_{t-1}) = \boldsymbol{z}_{t-1} - \mu_{t-1}$$
$$= \frac{1}{M} \begin{pmatrix} M-1 & -1 & \cdots & -1 \\ -1 & M-1 & \cdots & -1 \\ \cdots & \cdots & \cdots & \cdots \\ -1 & -1 & \cdots & M-1 \end{pmatrix} \boldsymbol{z}_{t-1} = \boldsymbol{M}\boldsymbol{z}_{t-1}. \tag{9}$$

As Remark 1 points out, all results about the tractability of the dendPLRNN also hold for the mean-centred dendPLRNN.

**State clipping** Since the ReLU function used in the dendPLRNN is non-saturating, states may diverge to infinity. As Theorem 2 guarantees, there is a simple and natural way to construct a "clipped" dendPLRNN

$$\boldsymbol{z}_t \;=\; \boldsymbol{A}\boldsymbol{z}_{t-1} + \boldsymbol{W} \sum_{b=1}^{B} \alpha_b \big[\max(0, \boldsymbol{z}_{t-1} - \boldsymbol{h}_b) - \max(0, \boldsymbol{z}_{t-1})\big] + \boldsymbol{h}_0. \tag{10}$$

Note that the results of Theorem 2 also hold true when the manifold attractor regularization is applied. This is detailed in Proposition 2 further below.

**Approximate posterior and hyperparameter settings** To estimate the true unknown posterior $p(\boldsymbol{z}|\boldsymbol{x})$, we make a Gaussian assumption for the approximate posterior $q_\phi(\boldsymbol{z}|\boldsymbol{x}) = \mathcal{N}(\boldsymbol{\mu}_\phi(\boldsymbol{x}), \boldsymbol{\Sigma}_\phi(\boldsymbol{x}))$, where mean and covariance are functions of the observations. Without any simplifying assumptions, the number of parameters in $\boldsymbol{\Sigma}_\phi(\boldsymbol{x}) \in \mathbb{R}^{MT \times MT}$ would scale unacceptably with time series length $T$. We therefore made a mean field assumption and factorized $q_\phi(\boldsymbol{z}|\boldsymbol{x})$ across time. Specifically, a time-dependent mean $\boldsymbol{\mu}_{t,\phi}$ and covariance $\boldsymbol{\Sigma}_{t,\phi}$ were parameterized through stacked convolutional networks which take the observations $\{\boldsymbol{x}_{t-w}...\boldsymbol{x}_{t+w}\}$ as inputs, with $w$ given by the largest kernel size. The mean is given by a 4-layer CNN with decreasing kernel sizes ($41, 31, 21$ and $11$, respectively), with the last layer of the CNN feeding into a fully connected layer. For the diagonal covariance, the observations are mapped directly onto the logarithms of the covariance through a single convolutional layer (with a kernel size of $41$) feeding into a fully connected layer. The classical motivation behind using CNNs rests on the assumption that the data contains translationally invariant patterns, and that this allows the recognition model to embed potentially meaningful temporal context into the latent representation (see e.g. Cui et al. (2016)). We note that while the mean-field approximation is computationally highly efficient, it makes potentially strongly simplifying assumptions (Blei et al., 2017; Bayer et al.) that may limit the ability of the encoder model to approximate the true posterior.

To train the dendPLRNN in the VI framework, Adam (Kingma & Ba, 2015), with a batch size of $1000$ and learning rate of $1e-3$ was used as the optimizer. For the training with BPTT, we used the Adam optimizer with an initial learning rate of $1e-3$ that was iteratively reduced during training down to $1e-5$. For each epoch we randomly sampled sequences of length $T_{seq} = 500$ (except for the Lorenz-63 runs, where $T_{seq} = 200$ time steps were sufficient) from the total training data pool of each dataset, which are then fed into the reconstruction method in batches of size 16. Initial weights were drawn from a uniform distribution.

To find optimal hyper-parameters we performed a grid search within $\lambda_{reg} \in \{0, 0.01, 0.1, 1, 10\}$ (VI), $\tau_{\text{TF}} \in \{1, 5, 10, 25, 50, 100\}$ (BPTT-TF), $M \in \{5, 10, 15, 20, 25, 30, 35, 40, 45, 50, 75, 100\}$, and $B \in \{0, 1, 2, 5, 10, 20, 35, 50\}$. Hyper-parameters chosen for the benchmarks in Sec. 4 are reported in Table S1 below (note that these may not fully agree with the ranges initially scanned, as given above, since we attempted to adjust them further in order to approximately match the number of parameters among models in Table S2).

Table S1: Hyperparameter settings for dendPLRNN VI/TF for the four different data sets from Sec. 4.

| Dataset | M | B | $M_{\text{reg}}/M$ | $\lambda_{\text{reg}}$ | $\tau_{\text{TF}}$ |
|---|---|---|---|---|---|
| Lorenz-63 | 22 | 20 | $1.0/-$ | $1.0/-$ | $-/25$ |
| Lorenz-96 | 42/50 | 50/30 | $1.0/-$ | $1.0/-$ | $-/10$ |
| Bursting Neuron | 26 | 50/47 | $0.5/-$ | $1.5/-$ | $-/5$ |
| Neural Population Model | 12/75 | 5/40 | $0.2/-$ | $1.0/-$ | $-/5$ |
| EEG | 117/128 | 50/50 | $0.8/0.5$ | $1.0/1.0$ | $-/10$ |

## 6.2 PERFORMANCE MEASURES

**Geometrical measure** $D_{\text{stsp}}$ used for evaluating attractor geometries (Fig. 4) measures the match between the ground truth distribution $p_{\text{true}}(\boldsymbol{x})$ and the generated distribution $p_{\text{gen}}(\boldsymbol{x} \mid \boldsymbol{z})$ through the

discrete binning approximation (Koppe et al., 2019)

$$D_{\text{stsp}}\left(p_{\text{true}}(\boldsymbol{x}), p_{\text{gen}}(\boldsymbol{x} \mid \boldsymbol{z})\right) \approx \sum_{k=1}^{K} \hat{p}_{\text{true}}^{(k)}(\boldsymbol{x}) \log \left( \frac{\hat{p}_{\text{true}}^{(k)}(\boldsymbol{x})}{\hat{p}_{\text{gen}}^{(k)}(\boldsymbol{x} \mid \boldsymbol{z})} \right), \tag{11}$$

where $K$ is the total number of bins, and $\hat{p}_{\text{true}}^{(k)}(\boldsymbol{x})$ and $\hat{p}_{\text{gen}}^{(k)}(\boldsymbol{x} \mid \boldsymbol{z})$ are estimated as relative frequencies through sampling trajectories from the benchmark DS and the trained reconstruction method, respectively. A range of $2\times$ the data standard deviation on each dimension was partioned into $m$ bins, yielding a total of $K = m^N$ bins, where $N$ is the dimension of the ground truth system. Initial transients are removed from sampled trajectories to ensure that the system has reached a limiting set. If the bin size is chosen too large, important geometrical details may be lost, while if it is chosen too small, noise and (low) sampling artifacts with many empty bins may misguide the approximation above. Here we chose a bin number of $m = 30$ per dimension as an optimal compromise that distinguished well between successful and poor reconstructions.

Since the number of bins needed to cover the relevant (populated) region of state space scales exponentially with the number of dimensions, for high-dimensional systems evaluating $D_{\text{stsp}}$ as outlined above is not feasible. We therefore resorted to an approximation of the densities based on Gaussian Mixture Models (GMMs), similar to a strategy outlined in (Koppe et al., 2019). Specifically, we approximate the true data distribution by a GMM $p_{\text{true}}(\boldsymbol{x}) \approx \frac{1}{T} \sum_{t=1}^{T} \mathcal{N}(\boldsymbol{x}_t, \boldsymbol{\Sigma})$ with Gaussians centered on the observed data points $\{\boldsymbol{x}_t\}$ and covariance $\boldsymbol{\Sigma}$, which we treat as a hyper-parameter that determines the granularity of the spatial resolution (similar to the bin size in Eq. 11). We can generate a likewise distribution by sampling trajectories from the trained models (or one very long trajectory) and placing Gaussians on the sampled data points, $p_{\text{gen}}(\boldsymbol{x}|\boldsymbol{z}) \approx \frac{1}{L} \sum_{l=1}^{L} \mathcal{N}(\hat{\boldsymbol{x}}_l \mid \boldsymbol{z}_l, \boldsymbol{\Sigma})$ (in the case of VI, rather than sampling, one could also use the model's distributional assumptions to build this posterior across the observations). For the Kullback-Leibler divergence between two GMMs efficient approximations are at hand (Hershey & Olsen, 2007). Here we employ a Monte Carlo approximation

$$\widetilde{D}_{\text{stsp}}\left(p_{true}(\boldsymbol{x}), p_{\text{gen}}(\boldsymbol{x}|\boldsymbol{z})\right) \approx \frac{1}{n} \sum_{i=1}^{n} \log \frac{1/T \sum_{t=1}^{T} \mathcal{N}(\boldsymbol{x}^{(i)}; \boldsymbol{x}_t, \boldsymbol{\Sigma})}{1/L \sum_{l=1}^{L} \mathcal{N}(\boldsymbol{x}^{(i)}; \hat{\boldsymbol{x}}_l, \boldsymbol{\Sigma})}, \tag{12}$$

where $n$ Monte-Carlo samples $\boldsymbol{x}^{(i)}$ are drawn from the GMM representing $p_{\text{true}}$. In practice, we set the covariance $\boldsymbol{\Sigma} = \sigma^2 \boldsymbol{I}$ equal to a scaled identity matrix, with a single hyperparamter $\sigma^2$. Scanning the range $\sigma^2 \in \{0.01, 0.02, 0.05, 0.1, 0.2, 0.5, 1, 2, 5\}$, we found that values for $\sigma^2 = 0.1 - 1.0$ to differentiate best between good and bad reconstructions. We chose $\sigma^2 = 1.0$ for numerical stability. For this setting, $D_{\text{stsp}}$ as derived with the binning method and $\widetilde{D}_{\text{stsp}}$ computed through the GMMs also correlated highly on the low-dimensional benchmark systems (see Figure S1).

**Power Spectrum Correlation**   The power spectrum correlations (PSC) were obtained by first sampling time series of 100,000 time steps, standardizing these, and computing dimension-wise Fast Fourier Transforms (using `scipy.fft`) for both the ground truth systems and model-simulated time series. Individual power spectra were then slightly smoothed with a Gaussian kernel, normalized, and the long, high-frequency tails of the spectra, mainly representing noise, were cut off. Smoothing width $\sigma$ and cutoff values scale linearly with the length of the time series used to compute the spectrum, and were chosen by visual inspection of the individual spectra. Dimension-wise correlations between smoothed power spectra were then averaged to obtain the reported PSC scores.

**Mean Squared Prediction Error**   A mean squared prediction error (PE) was computed across test sets of length $T = 1000$ by initializing the reconstruction model with the benchmark time series up to some time point $t$, from where it was then iterated forward by $n$ time steps to yield a prediction at time step $t + n$. The $n$-step ahead prediction error (PE) is then defined as the MSE between predicted and true observations:

$$PE(n) = \frac{1}{(T-n)} \sum_{t=1}^{T-n} ||x_{t+n} - \hat{x}_{t+n}||_2^2. \tag{13}$$

Note that for a chaotic system initially close trajectories will exponentially diverge, such that PEs for too large prediction steps $n$ are not meaningful anymore (in a chaotic system with noise, for large $n$

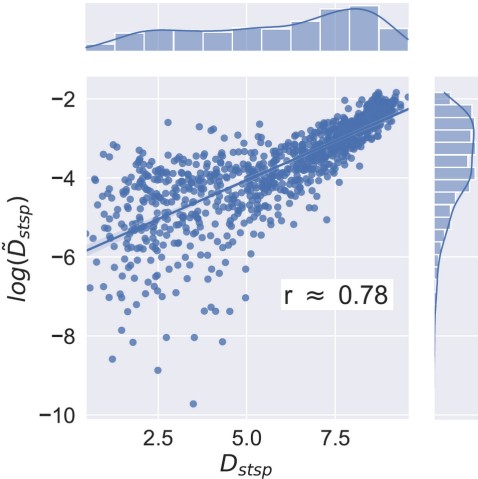

Figure S1: Correlation between the binning approximation ($m = 30$) and the logarithm of the GMM approximation ($\sigma^2 = 1$) to $D_{stsp}$ on the Lorenz-63 system for different noise realizations and variances. Similar as reported for the $KL_{\mathbf{z}}$ approximation in Koppe et al. (2019) we observed a logarithmic relation between the GMM and binning based measures.

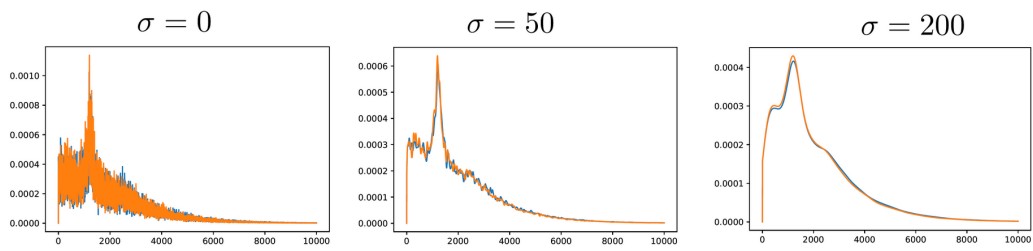

Figure S2: Example power spectrum for different values of the smoothing factor $\sigma^2$.

the PE may be high even when estimated from two different runs of the same ground truth model from the same initial condition; see (Koppe et al., 2019)). How quickly this happens depends on the rate of exponential divergence as quantified through the system's maximal Lyapunov exponent (Kantz & Schreiber, 2004).

## 6.3 FURTHER RESULTS

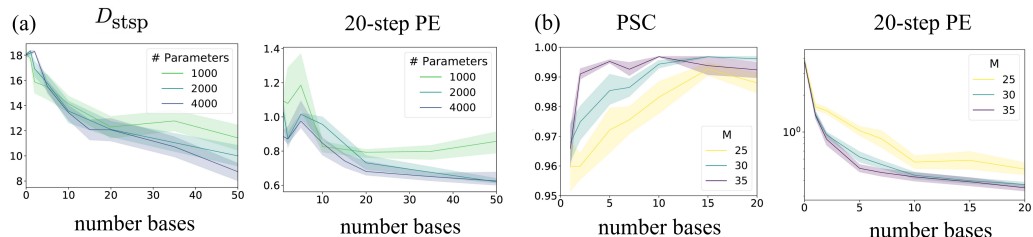

Figure S3: Effect of basis expansion for dendPLRNN trained by VI. (a) Agreement in attractor geometries (left) and 20-step ahead prediction error (right) for the bursting neuron system as a function of the number of bases ($B$) for fixed numbers of total parameters. (b) Agreement in power spectrum correlation (left) and 20-step ahead prediction error (right) for the Lorenz-96 system as a function of the number of bases ($B$) for different numbers of latent states ($M$).

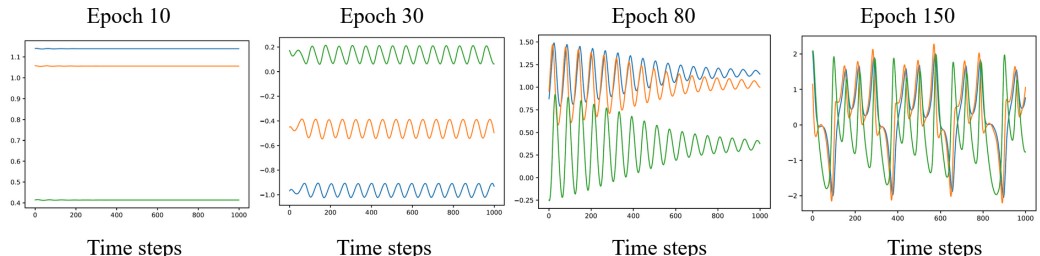

| Epoch 10 | Epoch 30 | Epoch 80 | Epoch 150 |
|---|---|---|---|
| Time steps | Time steps | Time steps | Time steps |

Figure S4: We observed that the dendPLRNN frequently underwent bifurcations between fixed point or various cyclic solutions until it reaches the chaotic behavior of the ground truth system.

Table S2: Comparison of dendPLRNN (Ours) trained by VI or BPTT+TF, and a standard PLRNN (Schmidt et al., 2021), trained by VI or BPTT+TF on four DS benchmarks (top) and three challenging data situations (bottom). Values are mean ± SEM.

| Dataset / Setting | Method | PSC | $D_{stsp}$ | 20-step PE | Dynamical variables | Parameters |
|---|---|---|---|---|---|---|
| Lorenz | dendPLRNN VI | **0.997 ± 0.001** | 0.80 ± 0.25 | 2.1e−3 ± 0.2e−3 | 22 | 1032 |
| | dendPLRNN TF | **0.997 ± 0.002** | **0.13 ± 0.18** | **9.2e−5 ± 2.8e−5** | 22 | 1032 |
| | PLRNN VI | 0.94 ± 0.004 | 16.6 ± 0.4 | 1.8e−1 ± 0.1e−1 | 22 | 1032 |
| | PLRNN TF | 0.994 ± 0.001 | 0.4 ± 0.09 | 4.3e−3 ± 0.2e−3 | 30 | 1011 |
| Lorenz-96 | dendPLRNN VI | 0.987 ± 0.001 | 0.10 ± 0.01 | 3.1e−1 ± 0.9e−1 | 42 | 4384 |
| | dendPLRNN TF | **0.998 ± 0.0001** | **0.04 ± 0.01** | 4.1e−2 ± 0.8e−2 | 50 | 4480 |
| | PLRNN VI | 0.93 ± 0.002 | 1.68 ± 0.03 | **2.1e−3 ± 0.2e−3** | 60 | 4260 |
| | PLRNN TF | 0.996 ± 0.0003 | 0.05 ± 0.01 | 2.2e−1 ± 0.2e−1 | 64 | 4700 |
| Bursting Neuron | dendPLRNN VI | 0.55 ± 0.03 | 7.5 ± 0.4 | 6.1e−1 ± 0.1e−1 | 26 | 2052 |
| | dendPLRNN TF | **0.76 ± 0.04** | **2.9 ± 1.3** | **6.1e−2 ± 2.2e−2** | 26 | 2040 |
| | PLRNN VI | 0.54 ± 0.01 | 17.5 ± 0.5 | 1.17 ± 0.14 | 42 | 2021 |
| | PLRNN TF | 0.72 ± 0.07 | 3.2 ± 2.0 | 7.0e−2 ± 2.7e−2 | 43 | 2021 |
| Neural Population Model | dendPLRNN VI | 0.45 ± 0.05 | 0.56 ± 0.05 | 0.82 ± 0.09 | 12 | 821 |
| | dendPLRNN TF | **0.51 ± 0.01** | **0.19 ± 0.02** | 1.53 ± 0.03 | 75 | 9990 |
| | PLRNN VI | 0.48 ± 0.01 | 11.65 ± 1.32 | **0.68 ± 0.09** | 13 | 832 |
| | PLRNN TF | 0.38 ± 0.15 | 9.6 ± 7.3 | 19 ± 23 | 98 | 12102 |
| Low amount of data | dendPLRNN VI | 0.967 ± 0.007 | **4.36 ± 0.10** | 2.8e−2 ± 0.2e−2 | 22 | 1032 |
| | dendPLRNN TF | **0.97 ± 0.04** | 6.9 ± 5.3 | **1.5e−2 ± 0.9e−2** | 22 | 1032 |
| | PLRNN VI | 0.96 ± 0.01 | 18.1 ± 0.10 | 1.08 ± 0.02 | 30 | 1020 |
| | PLRNN TF | 0.96 ± 0.04 | 9.0 ± 5.4 | 1.8e−2 ± 0.5e−2 | 30 | 1011 |
| Partially observed | dendPLRNN VI | 0.940 ± 0.006 | 12.6 ± 1.0 | 6.5e−2 ± 1.4e−2 | 22 | 1032 |
| | dendPLRNN TF | 0.993 ± 0.003 | **0.54 ± 0.16** | 5.3e−3 ± 0.2e−3 | 22 | 1032 |
| | PLRNN VI | 0.944 ± 0.002 | 17.2 ± 0.2 | 2.7e−1 ± 0.03e−1 | 30 | 1020 |
| | PLRNN TF | **0.994 ± 0.003** | 0.56 ± 0.34 | **5.0e−3 ± 0.2e−3** | 30 | 1011 |
| High noise | dendPLRNN VI | 0.973 ± 0.006 | 4.9 ± 0.75 | 3.5e−2 ± 0.1e−2 | 22 | 1032 |
| | dendPLRNN TF | **0.995 ± 0.002** | **0.4 ± 0.13** | 4.6e−3 ± 0.4e−3 | 22 | 1032 |
| | PLRNN VI | 0.94 ± 0.004 | 18.2 ± 0.04 | 6.4e−1 ± 0.1e−1 | 30 | 1020 |
| | PLRNN TF | 0.994 ± 0.002 | 0.5 ± 0.08 | **4.3e−3 ± 0.2e−3** | 22 | 1032 |

**EEG Dataset** Electroencephalogram (EEG) data were taken from a study by (Schalk et al., 2000) available at https://physionet.org/content/eegmmidb/1.0.0/. These are 64-channel EEG data obtained from human subjects during different motor and imagery tasks. We trained the dendPLRNN using BPTT+TF on the "eyes open" baseline time series from subject 0, which had a total of 9760 time steps. The signal was standardized and smoothed with a Hann function, using numpy.hanning and a window length of 15. Results for ground-truth and freely generated EEG signals from several brain regions are shown in figure S5.

## 6.4 DETAILS ON DYNAMICAL SYSTEMS BENCHMARKS

**Lorenz-63 system** The famous 3d chaotic Lorenz attractor with the butterfly wing shape, originally proposed in (Lorenz, 1963), is defined by

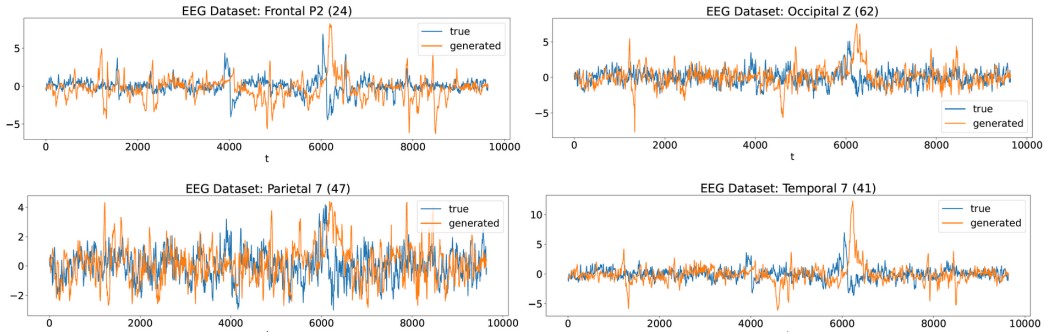

Figure S5: EEG recordings from frontal, occipital, parietal and temporal lobe vs. freely generated trajectories, sampled from the dendPLRNN, trained with BPTT ($M = 128, B = 50, \tau = 10, M_{\text{reg}}/M = 0.5, \lambda = 1.0$).

$$\frac{dx}{dt} = \sigma(y - x) + \frac{d\epsilon_1(t)}{dt},$$
$$\frac{\mathrm{d}y}{\mathrm{d}t} = x(\rho - z) - y + \frac{d\epsilon_2(t)}{dt}, \tag{14}$$
$$\frac{\mathrm{d}z}{\mathrm{d}t} = xy - \beta z + \frac{d\epsilon_3(t)}{dt}.$$

Parameters used for producing ground truth data in the chaotic regime were $\sigma = 10, \rho = 28$, and $\beta = 8/3$. Process noise was injected into the system by drawing from a Gaussian term $d\epsilon \sim \mathcal{N}(\mathbf{0}, 0.01dt \times \mathbf{I})$.

**Bursting neuron model**    The 3d biophysical bursting neuron model was introduced in (Durstewitz, 2009) and is defined by one voltage and two ion channel gating variables (one slow and one fast):

$$-C_m\dot{V} = g_L\left(V - E_L\right) + g_{Na}m_\infty(V)\left(V - E_{Na}\right)$$
$$+ g_K n\left(V - E_K\right) + g_M h\left(V - E_K\right) \tag{15}$$
$$+ g_{NMDA}\left[1 + .33e^{-.0625V}\right]^{-1}\left(V - E_{NMDA}\right)$$

$$\dot{h} = \frac{h_\infty(V) - h}{\tau_h}$$
$$\dot{n} = \frac{n_\infty(V) - n}{\tau_n} \tag{16}$$

The limiting values of the ionic gates are given by

$$\{m_\infty, n_\infty, h_\infty\} = \left[1 + e^{(\{V_{hNa}, V_{hK}, V_{hM}\} - V)/\{k_{Na}, k_K, k_M\}}\right]^{-1}. \tag{17}$$

We borrowed parameter settings from Schmidt et al. (2021) to place the system into the burst-firing regime:

$$C_m = 6\mu\text{F}, g_L = 8\text{mS}, E_L = -80\text{mV}, g_{Na} = 20\text{mS}, E_{Na} = 60\text{mV}, V_{hNa} = -20\text{mV},$$
$$k_{Na} = 15, g_K = 10\text{mS}, E_K = -90\text{mV}, V_{hK} = -25\text{mV}, k_K = 5, \tau_n = 1 \text{ ms}, g_M = 25\text{mS}$$
$$V_{hM} = -15\text{mV}, k_M = 5, \tau_h = 200 \text{ ms}, g_{NMDA} = 10.2\text{mS}$$

**Lorenz-96 system** The Lorenz-96 is a high-dimensional, spatially extended weather model, also introduced by Edward Lorenz (Lorenz, 1996):

$$\frac{dx_i}{dt} = (x_{i+1} - x_{i-2})x_{i-1} - x_i + F + +\frac{d\epsilon_i(t)}{dt}, \; i = 1 \ldots N, \tag{18}$$

with (constant) forcing term $F$. $F = 8$ is a common choice that leads to chaotic behavior. Process noise was added as for the Lorenz-63 system, $d\epsilon \sim \mathcal{N}(\mathbf{0}, 0.01dt \times \mathbf{I})$. In our simulations we used $N = 10$, but in principle the system allows for arbitrary dimensionality.

**Neural population model** A larger-scale neural population model was recently introduced in Landau & Sompolinsky (2018) to examine the effect of structured connectivity on top of a randomly initialized network matrix. Specifically, an independently Gaussian distributed weight structure was combined with a rank-1 component with coupling strength $J_1$. The dynamics of the single unit currents were defined as

$$\frac{d\mathbf{h}}{dt} = -\mathbf{h} + \mathbf{J}\phi(\mathbf{h}) + \frac{J_1}{\sqrt{N}}\xi v^T \phi(\mathbf{h}), \tag{19}$$

where $\phi(\mathbf{h}) = \tanh(\mathbf{h}(t))$. We produced a 50-dimensional chaotic network model based on the code provided in Landau & Sompolinsky (2018) using $J_1 = 0.09$ and seeding the random number generator with 35.

The Lorenz-63 and Lorenz-96 systems were simulated using `scipy.integrate`, while for the bursting neuron and neural population model we used the code provided in Schmidt et al. (2021) and Landau & Sompolinsky (2018), respectively.

## 6.5 THEORETICAL ANALYSIS

Consider the PLRNN with linear spline basis expansion as defined by Eq. 1, Eq. 4, reproduced here for convenience:

$$\boldsymbol{z}_t = \boldsymbol{A}\boldsymbol{z}_{t-1} + \boldsymbol{W}\sum_{b=1}^{B}\alpha_b \max(0, \boldsymbol{z}_{t-1} - \boldsymbol{h}_b) + \boldsymbol{h}_0 + \boldsymbol{C}\boldsymbol{s}_t + \boldsymbol{\epsilon}_t, \tag{20}$$

where $\boldsymbol{\epsilon}_t \sim N(0, \boldsymbol{\Sigma})$, $E[\boldsymbol{\epsilon}_t, \boldsymbol{\epsilon}_{t'}^\mathsf{T}] = 0$ for $t \neq t'$, $\alpha_b \in \mathbb{R}$ are scalar weighting factors and $\boldsymbol{h}_b \in \mathbb{R}^M$ different ReLU "activation thresholds", and all other parameters are as in conventional PLRNNs (Koppe et al., 2019).

Defining

$$\boldsymbol{D}_{\Omega(t-1)}^{(b)}(\boldsymbol{z}_{t-1} - \boldsymbol{h}_b) := \max(0, \boldsymbol{z}_{t-1} - \boldsymbol{h}_b), \tag{21}$$

Eq. 20 can be rewritten as

$$\boldsymbol{z}_t = \left(\boldsymbol{A} + \boldsymbol{W}\sum_{b=1}^{B}\alpha_b \, \boldsymbol{D}_{\Omega(t-1)}^{(b)}\right)\boldsymbol{z}_{t-1}$$

$$+ \, \boldsymbol{W}\sum_{b=1}^{B}\alpha_b \, \boldsymbol{D}_{\Omega(t-1)}^{(b)}(-\boldsymbol{h}_b) \, + \, \boldsymbol{h}_0 \, + \, \boldsymbol{C}\boldsymbol{s}_t + \boldsymbol{\epsilon}_t, \tag{22}$$

where $\boldsymbol{D}_{\Omega(t-1)}^{(b)} = \text{diag}\big(d_{1,t-1}^{(b)}, d_{2,t-1}^{(b)}, \cdots, d_{M,t-1}^{(b)}\big)$ are diagonal binary indicator matrices with $d_{m,t-1}^{(b)} = 1$ if $z_{m,t-1} > h_{m,b}$ and 0 otherwise.

Defining

$$\boldsymbol{D}_{\Omega(t-1)}^{B} := \sum_{b=1}^{B} \alpha_b \, \boldsymbol{D}_{\Omega(t-1)}^{(b)},$$

$$\boldsymbol{h}_{\Omega(t-1)}^{B} := \sum_{b=1}^{B} \alpha_b \, \boldsymbol{D}_{\Omega(t-1)}^{(b)}(-\boldsymbol{h}_b), \tag{23}$$

$$\boldsymbol{W}_{\Omega(t-1)}^{B} := \boldsymbol{A} + \boldsymbol{W} \, \boldsymbol{D}_{\Omega(t-1)}^{B},$$

and considering the autonomous system (i.e., without external inputs or noise terms), Eq. 22 can be rewritten as

$$\boldsymbol{z}_t = \boldsymbol{W}_{\Omega(t-1)}^{B} \, \boldsymbol{z}_{t-1} + \boldsymbol{W} \, \boldsymbol{h}_{\Omega(t-1)}^{B} + \boldsymbol{h}_0. \tag{24}$$

Fixed points and cycles of Eq. 20, and their eigenvalue spectra, can now be computed in a way analogous to standard PLRNNs. Specifically, solving the equation $F(\boldsymbol{z}^{*1}) = \boldsymbol{z}^{*1}$, fixed points of the dendPLRNN are given by

$$\boldsymbol{z}^{*1} = \left(\boldsymbol{I} - \boldsymbol{W}_{\Omega(t^{*1})}^{B}\right)^{-1}\left[\boldsymbol{W} \, \boldsymbol{h}_{\Omega(t^{*1})}^{B} + \boldsymbol{h}_0\right], \tag{25}$$

where $\boldsymbol{z}^{*1} = \boldsymbol{z}_{t^{*1}} = \boldsymbol{z}_{t^{*1}-1}$, and $\det(\boldsymbol{I} - \boldsymbol{W}_{\Omega(t^{*1})}^{B}) = P_{\boldsymbol{W}_{\Omega(t^{*1})}^{B}}(1) \neq 0$, i.e. $\boldsymbol{W}_{\Omega(t^{*1})}^{B}$ has no eigenvalue equal to 1.

For $n > 1$, an $n$-cycle with periodic points $\{\boldsymbol{z}^{*n}, F(\boldsymbol{z}^{*n}), F^2(\boldsymbol{z}^{*n}), \cdots, F^{n-1}(\boldsymbol{z}^{*n})\}$ of map $F$ can be obtained by solving $F^n(\boldsymbol{z}^{*n}) = \boldsymbol{z}^{*n}$. Therefore, in order to find the periodic points, we first compute $F^n$ in the following way:

$$\boldsymbol{z}_t = F(\boldsymbol{z}_{t-1}) = \boldsymbol{W}_{\Omega(t-1)}^{B} \, \boldsymbol{z}_{t-1} + \boldsymbol{W} \, \boldsymbol{h}_{\Omega(t-1)}^{B} + \boldsymbol{h}_0,$$

$$\boldsymbol{z}_{t+1} = F^2(\boldsymbol{z}_{t-1}) = F(\boldsymbol{z}_t) = \boldsymbol{W}_{\Omega(t)}^{B} \boldsymbol{W}_{\Omega(t-1)}^{B} \, \boldsymbol{z}_{t-1} + \left(\boldsymbol{W}_{\Omega(t)}^{B} \, \boldsymbol{W} \, \boldsymbol{h}_{\Omega(t-1)}^{B} + \boldsymbol{W} \, \boldsymbol{h}_{\Omega(t)}^{B}\right)$$
$$+ \left(\boldsymbol{W}_{\Omega(t)}^{B} + \boldsymbol{I}\right)\boldsymbol{h}_0,$$

$$\boldsymbol{z}_{t+2} = F^3(\boldsymbol{z}_{t-1}) = F(\boldsymbol{z}_{t+1}) = \boldsymbol{W}_{\Omega(t+1)}^{B} \boldsymbol{W}_{\Omega(t)}^{B} \boldsymbol{W}_{\Omega(t-1)}^{B} \, \boldsymbol{z}_{t-1} + \left(\boldsymbol{W}_{\Omega(t+1)}^{B} \boldsymbol{W}_{\Omega(t)}^{B} \boldsymbol{W} \boldsymbol{h}_{\Omega(t-1)}^{B}\right.$$
$$\left. + \boldsymbol{W}_{\Omega(t+1)}^{B} \boldsymbol{W} \boldsymbol{h}_{\Omega(t)}^{B} + \boldsymbol{W} \boldsymbol{h}_{\Omega(t+1)}^{B}\right) + \left(\boldsymbol{W}_{\Omega(t+1)}^{B} \boldsymbol{W}_{\Omega(t)}^{B} + \boldsymbol{W}_{\Omega(t+1)}^{B} + \boldsymbol{I}\right)\boldsymbol{h}_0,$$

$$\vdots$$

$$\boldsymbol{z}_{t+(n-1)} = F^n(\boldsymbol{z}_{t-1}) = \prod_{i=2}^{n+1} \boldsymbol{W}_{\Omega(t+n-i)}^{B} \, \boldsymbol{z}_{t-1} + \sum_{j=2}^{n}\left[\prod_{i=2}^{n-j+2} \boldsymbol{W}_{\Omega(t+n-i)}^{B} \, \boldsymbol{W} \, \boldsymbol{h}_{\Omega(t+j-3)}^{B}\right]$$

$$+ \boldsymbol{W} \boldsymbol{h}_{\Omega(t+n-2)}^{B} + \left(\sum_{j=2}^{n}\prod_{i=2}^{n-j+2} \boldsymbol{W}_{\Omega(t+n-i)}^{B} + \boldsymbol{I}\right)\boldsymbol{h}_0, \tag{26}$$

where

$$\prod_{i=2}^{n+1} \boldsymbol{W}_{\Omega(t+n-i)}^{B} = \boldsymbol{W}_{\Omega(t+n-2)}^{B} \boldsymbol{W}_{\Omega(t+n-3)}^{B} \cdots \boldsymbol{W}_{\Omega(t-1)}^{B}.$$

Defining $t + n - 1 =: t^{*n}$, the periodic point $\boldsymbol{z}^{*n}$ of the $n$-cycle of $F$ can now be obtained as the fixed point of the $n$-times iterated map $F^n$ as

$$\boldsymbol{z}^{*n} = \left(\boldsymbol{I} - \prod_{i=1}^{n} \boldsymbol{W}_{\Omega(t^{*n}-i)}^{B}\right)^{-1}\left(\sum_{j=2}^{n}\left[\prod_{i=1}^{n-j+1} \boldsymbol{W}_{\Omega(t^{*n}-i)}^{B} \boldsymbol{W} \boldsymbol{h}_{\Omega(t^{*n}-n+j-2)}^{B}\right] + \boldsymbol{W} \boldsymbol{h}_{\Omega(t^{*n}-1)}^{B}\right.$$

$$+ \Big( \sum_{j=2}^{n} \prod_{i=1}^{n-j+1} \boldsymbol{W}^{B}_{\Omega(t^{*n}-i)} + \boldsymbol{I} \Big) \boldsymbol{h}_0 \Big), \tag{27}$$

where $\boldsymbol{z}^{*n} = \boldsymbol{z}_{t^n} = \boldsymbol{z}_{t^{*n}-n}$, if $(\boldsymbol{I} - \prod_{i=1}^{n} \boldsymbol{W}^{B}_{\Omega(t^{*n}-i)})$ is invertible, i.e.

$$\det \Big( \boldsymbol{I} - \prod_{i=1}^{n} \boldsymbol{W}^{B}_{\Omega(t^{*n}-i)} \Big) = P_{\prod_{i=1}^{n} \boldsymbol{W}^{B}_{\Omega(t^{*n}-i)}}(1) \neq 0,$$

which implies $\boldsymbol{W}_{\Omega^{*n}} := \prod_{i=1}^{n} \boldsymbol{W}^{B}_{\Omega(t^{*n}-i)}$ has no eigenvalue equal to 1.

**Remark 1.** *These results about fixed points and $n$-cycles also hold for the mean-centred dendPLRNN. This can easily be seen by defining $\boldsymbol{W}^{B}_{\Omega(t-1)} := \boldsymbol{A} + \boldsymbol{W}\,\boldsymbol{D}^{B}_{\Omega(t-1)}\,\boldsymbol{M}$ and noting that the elements of $\boldsymbol{D}^{(b)}_{\Omega(t-1)}$ are now determined by the mean-centred latent states. That is $d^{(b)}_{m,t-1} = 1$ if $z_{m,t-1} - \frac{1}{M} \sum_{j=1}^{M} z_{j,t-1} > h_{m,b}$ and 0 otherwise. The rest of the calculations then proceeds as above.*

### 6.5.2 SUB-REGIONS AND DISCONTINUITY BOUNDARIES CORRESPONDING TO SYSTEM EQ. 22

Consider system Eq. 22 without external input and noise terms. Denoting $\boldsymbol{h}_b = (h_{1,b}, h_{2,b}, \cdots, h_{M,b})^\mathsf{T}$ in Eq. 22, for $b = 1, 2, \cdots, B$, we can order the elements $h_{j,1}, h_{j,2}, \cdots, h_{j,B}$ for every $j \in \{1, 2, \cdots, M\}$. Without loss of generality, let

$$h_{j,1} < h_{j,2} < \cdots < h_{j,B}, \quad j = 1, 2, \cdots, M. \tag{28}$$

Then, for every $j$, we define the intervals $I_{j,b}$ as follows:

$$I_{j,1} := (-\infty, h_{j,1}],$$
$$I_{j,b} := (h_{j,b-1}, h_{j,b}], \quad b = 2, 3, \cdots, B, \tag{29}$$
$$I_{j,B+1} := (h_{j,B}, +\infty).$$

By definition of $\boldsymbol{D}^{(i)}_{\Omega(t-1)}$ in Eq. 22, the phase space is separated into $(B+1)^M$ sub-regions by $MB(B+1)^{M-1}$ hyper-surfaces as discontinuity boundaries. Every sub-region can be defined by the thresholds $\boldsymbol{h}_b$ as Cartesian product of suitable intervals in Eq. 29 for $j \in \{1, 2, \cdots, M\}$. (Note that if in Eq. 28 we had " $\leq$ " instead of strict inequalities " $<$ ", obviously the number of intervals, hence sub-regions, would decrease.) In each sub-region the matrices $\boldsymbol{D}^{(b)}_{\Omega(t-1)}$, $b = 1, 2, \cdots, B$, have a different configuration. Therefore, in Eq. 24 there are $(B+1)^M$ different forms for $\boldsymbol{D}^{B}_{\Omega(t-1)}$, and so for $\boldsymbol{W}^{B}_{\Omega(t-1)}$ and $\boldsymbol{h}^{B}_{\Omega(t-1)}$ as well. Hence, indexing $\boldsymbol{D}^{B}_{\Omega(t-1)}$, $\boldsymbol{W}^{B}_{\Omega(t-1)}$ and $\boldsymbol{h}^{B}_{\Omega(t-1)}$ as $\boldsymbol{D}^{B}_{(r)}$, $\boldsymbol{W}^{B}_{(r)}$ and $\boldsymbol{h}^{B}_{(r)}$ for $r \in \{1, 2, \cdots, (B+1)^M\}$, Eq. 22 can be written as

$$\boldsymbol{z}_t = \boldsymbol{W}^{B}_{(r)}\,\boldsymbol{z}_{t-1} + \boldsymbol{W}\,\boldsymbol{h}^{B}_{(r)} + \boldsymbol{h}_0. \tag{30}$$

To visualize the sub-regions and their borders, let for example $M = 2$ and $B = 2$. In this case there are 9 sub-regions divided by 12 borders. As illustrated in Fig. 6.5.2, there are different matrices $\boldsymbol{D}^{(b)}_{\Omega(t-1)}$, $b = 1, 2$, and $\boldsymbol{D}^{B}_{(r)} = \boldsymbol{D}^{2}_{(r)}$, $r = 1, 2, \cdots, 9$, for each sub-region.

### 6.5.3 BOUNDED ORBITS ARE COMPATIBLE WITH THE MANIFOLD ATTRACTOR REGULARIZATION

**Proposition 2.** *The results of Theorem 2 are also true when the manifold-attractor regularization, Eq. 6, is strictly enforced for the dendPLRNN, Eq. 10.*

*Proof.* Assume $\boldsymbol{A}$, $\boldsymbol{W}$, $\tilde{\phi}(z_{t-1})$ (see proof of Theorem 2 in Appx. 6.5.6 for the definition) and $\boldsymbol{h}_0$ have the partitioned forms

$$\boldsymbol{A} = \left( \begin{array}{c|c} \boldsymbol{I}_{reg} & \boldsymbol{O}^\mathsf{T} \\ \hline \boldsymbol{O} & \boldsymbol{A}_{nreg} \end{array} \right), \qquad \boldsymbol{W} = \left( \begin{array}{c|c} \boldsymbol{O}_{reg} & \boldsymbol{O}^\mathsf{T} \\ \hline \boldsymbol{S} & \boldsymbol{W}_{nreg} \end{array} \right),$$

| $z_2 = h_{2,2}$ | $z_2 = h_{2,1}$ | |
|---|---|---|
| $D^{(1)}_{\Omega(t-1)} = \begin{pmatrix} 0 & 0 \\ 0 & 1 \end{pmatrix}$ $\quad D^{B}_{(1)} = \begin{pmatrix} 0 & 0 \\ 0 & \alpha_1+\alpha_2 \end{pmatrix}$ $\quad D^{(2)}_{\Omega(t-1)} = \begin{pmatrix} 0 & 0 \\ 0 & 1 \end{pmatrix}$ | $D^{(1)}_{\Omega(t-1)} = \begin{pmatrix} 1 & 0 \\ 0 & 1 \end{pmatrix}$ $\quad D^{B}_{(4)} = \begin{pmatrix} \alpha_1 & 0 \\ 0 & \alpha_1+\alpha_2 \end{pmatrix}$ $\quad D^{(2)}_{\Omega(t-1)} = \begin{pmatrix} 0 & 0 \\ 0 & 1 \end{pmatrix}$ | $D^{(1)}_{\Omega(t-1)} = \begin{pmatrix} 1 & 0 \\ 0 & 1 \end{pmatrix}$ $\quad D^{B}_{(7)} = \begin{pmatrix} \alpha_1+\alpha_2 & 0 \\ 0 & \alpha_1+\alpha_2 \end{pmatrix}$ $\quad D^{(2)}_{\Omega(t-1)} = \begin{pmatrix} 1 & 0 \\ 0 & 1 \end{pmatrix}$ |
| $D^{(1)}_{\Omega(t-1)} = \begin{pmatrix} 0 & 0 \\ 0 & 1 \end{pmatrix}$ $\quad D^{B}_{(2)} = \begin{pmatrix} 0 & 0 \\ 0 & \alpha_1 \end{pmatrix}$ $\quad D^{(2)}_{\Omega(t-1)} = \begin{pmatrix} 0 & 0 \\ 0 & 0 \end{pmatrix}$ | $D^{(1)}_{\Omega(t-1)} = \begin{pmatrix} 1 & 0 \\ 0 & 1 \end{pmatrix}$ $\quad D^{B}_{(5)} = \begin{pmatrix} \alpha_1 & 0 \\ 0 & \alpha_1 \end{pmatrix}$ $\quad D^{(2)}_{\Omega(t-1)} = \begin{pmatrix} 0 & 0 \\ 0 & 0 \end{pmatrix}$ | $D^{(1)}_{\Omega(t-1)} = \begin{pmatrix} 1 & 0 \\ 0 & 1 \end{pmatrix}$ $\quad D^{B}_{(8)} = \begin{pmatrix} \alpha_1+\alpha_2 & 0 \\ 0 & \alpha_1 \end{pmatrix}$ $\quad D^{(2)}_{\Omega(t-1)} = \begin{pmatrix} 1 & 0 \\ 0 & 0 \end{pmatrix}$ |
| $D^{(1)}_{\Omega(t-1)} = \begin{pmatrix} 0 & 0 \\ 0 & 0 \end{pmatrix}$ $\quad D^{B}_{(3)} = \begin{pmatrix} 0 & 0 \\ 0 & 0 \end{pmatrix}$ $\quad D^{(2)}_{\Omega(t-1)} = \begin{pmatrix} 0 & 0 \\ 0 & 0 \end{pmatrix}$ | $D^{(1)}_{\Omega(t-1)} = \begin{pmatrix} 1 & 0 \\ 0 & 0 \end{pmatrix}$ $\quad D^{B}_{(6)} = \begin{pmatrix} \alpha_1 & 0 \\ 0 & 0 \end{pmatrix}$ $\quad D^{(2)}_{\Omega(t-1)} = \begin{pmatrix} 0 & 0 \\ 0 & 0 \end{pmatrix}$ | $D^{(1)}_{\Omega(t-1)} = \begin{pmatrix} 1 & 0 \\ 0 & 0 \end{pmatrix}$ $\quad D^{B}_{(9)} = \begin{pmatrix} \alpha_1+\alpha_2 & 0 \\ 0 & 0 \end{pmatrix}$ $\quad D^{(2)}_{\Omega(t-1)} = \begin{pmatrix} 1 & 0 \\ 0 & 0 \end{pmatrix}$ |
| | $z_1 = h_{1,1}$ | $z_1 = h_{1,2}$ |

Figure S6: Example of different sub-regions and related matrices $\boldsymbol{D}^{(b)}_{\Omega(t-1)}, b = 1, 2$, and $\boldsymbol{D}^{B}_{(r)}, r = 1, 2, \cdots, 9$, for $M = 2$ and $B = 2$. Here, it is assumed that the components of $\boldsymbol{h}_1 = (h_{1,1}, h_{2,1})^{\mathsf{T}}$ and $\boldsymbol{h}_2 = (h_{1,2}, h_{2,2})^{\mathsf{T}}$ satisfy Eq. 28 with " $<$ ".

$$\boldsymbol{h}_0 = \left( \frac{\boldsymbol{h}_0^{reg}}{\boldsymbol{h}_0^{nreg}} \right), \qquad \tilde{\phi}(z_{t-1}) = \left( \frac{\tilde{\phi}_{reg}(z_{t-1})}{\tilde{\phi}_{nreg}(z_{t-1})} \right), \tag{31}$$

where $\boldsymbol{I}_{M_{reg} \times M_{reg}} =: \boldsymbol{I}_{reg} \in \mathbb{R}^{M_{reg} \times M_{reg}}, \boldsymbol{O}_{M_{reg} \times M_{reg}} =: \boldsymbol{O}_{reg} \in \mathbb{R}^{M_{reg} \times M_{reg}}, \boldsymbol{O}, \boldsymbol{S} \in \mathbb{R}^{(M-M_{reg}) \times M_{reg}}$, the sub-matrices $\boldsymbol{A}_{\{M_{reg}+1:M, M_{reg}+1:M\}} =: \boldsymbol{A}_{nreg} \in \mathbb{R}^{(M-M_{reg}) \times (M-M_{reg})}$ and $\boldsymbol{W}_{\{M_{reg}+1:M, M_{reg}+1:M\}} =: \boldsymbol{W}_{nreg} \in \mathbb{R}^{(M-M_{reg}) \times (M-M_{reg})}$ are diagonal and off-diagonal respectively. Furthermore, $\boldsymbol{h}_0^{reg}, \tilde{\phi}_{reg}(z_{t-1}) \in \mathbb{R}^{M_{reg}}$ and $\boldsymbol{h}_0^{\{M_{reg}+1:M, M_{reg}+1:M\}} =: \boldsymbol{h}_0^{nreg}$, $\tilde{\phi}_{\{M_{reg}+1:M\}}(z_{t-1}) =: \tilde{\phi}_{nreg}(z_{t-1}) \in \mathbb{R}^{M-M_{reg}}$.

In this case $\|\boldsymbol{A}\| = \sigma_{\max}(\boldsymbol{A}) = \max\{1, \sigma_{\max}(\boldsymbol{A}_{nreg})\}$ and

$$\left\| \boldsymbol{A}^j \boldsymbol{W} \tilde{\phi}(\boldsymbol{z}_{T-1-j}) \right\| = \left\| \left( \frac{\boldsymbol{O}}{\boldsymbol{A}_{neg}^j \boldsymbol{S} \tilde{\phi}_{nreg}(z_{t-1}) + \boldsymbol{A}_{neg}^j \boldsymbol{W}_{neg} \tilde{\phi}_{nreg}(z_{t-1})} \right) \right\|$$

$$= \left\| \boldsymbol{A}_{neg}^j \boldsymbol{S} \tilde{\phi}_{nreg}(z_{t-1}) + \boldsymbol{A}_{neg}^j \boldsymbol{W}_{neg} \tilde{\phi}_{nreg}(z_{t-1}) \right\|,$$

$$\left\| \boldsymbol{A}^j \boldsymbol{W} \boldsymbol{h}_0 \right\| = \left\| \left( \frac{\boldsymbol{O}}{\boldsymbol{A}_{neg}^j \boldsymbol{S} \boldsymbol{h}_0^{nreg} + \boldsymbol{A}_{neg}^j \boldsymbol{W}_{neg} \boldsymbol{h}_0^{nreg}} \right) \right\|$$

$$= \left\| \boldsymbol{A}_{neg}^j \boldsymbol{S} \boldsymbol{h}_0^{nreg} + \boldsymbol{A}_{neg}^j \boldsymbol{W}_{neg} \boldsymbol{h}_0^{nreg} \right\|. \tag{32}$$

Thus, for $\sigma_{\max}(\boldsymbol{A}_{nreg}) < 1$

$$\|\boldsymbol{z}_T\| \leq \|\boldsymbol{A}\|^{T-1} \|\boldsymbol{z}_1\| + \sum_{j=0}^{T-2} \left\| \boldsymbol{A}^j \boldsymbol{W} \tilde{\phi}(\boldsymbol{z}_{T-1-j}) \right\| + \sum_{j=0}^{T-2} \left\| \boldsymbol{A}^j \boldsymbol{h}_0 \right\|$$

$$\leq \|\boldsymbol{z}_1\| + \left( \tilde{c} + \|h_0\| \right) \left( \|\boldsymbol{S}\| + \|\boldsymbol{W}_{neg}\| \right) \sum_{j=0}^{T-2} \|\boldsymbol{A}_{neg}\|^j$$

$$= \frac{\big(\tilde{c} + \|h_0\|\big)\big(\|\boldsymbol{S}\| + \|\boldsymbol{W}_{neg}\|\big)}{1 - \|\boldsymbol{A}_{neg}\|} < \infty. \tag{33}$$

$\square$

### 6.5.4 PROOF OF PROPOSITION 1

*Proof.* For $\boldsymbol{A} = (a_{ij}) \in \mathbb{R}^{M \times M}$, $\boldsymbol{W} = (w_{ij}) \in \mathbb{R}^{M \times M}$, $\boldsymbol{\epsilon}_t = (\epsilon_{1,t}, \epsilon_{2,t}, \cdots, \epsilon_{M,t})^{\mathsf{T}}$, $\boldsymbol{s}_t = (s_{1,t}, s_{2,t}, \cdots, s_{M,t})^{\mathsf{T}}$ and $\boldsymbol{C} = (c_{ij}) \in \mathbb{R}^{M \times M}$, writing Eq. 22 in scalar form yields

$$z_{l,t} = \sum_{j=1}^{M} a_{lj} z_{j,t-1} + \sum_{j=1}^{M} w_{lj} \sum_{b=1}^{B} \alpha_b\, d_{j,t-1}^{(b)} [z_{j,t-1} - h_{j,b}] + h_{l,0} + \sum_{j=1}^{M} c_{lj}\, s_{j,t} + \epsilon_{l,t}$$

$$= \sum_{j=1}^{M} \left( a_{lj} z_{j,t-1} + w_{lj} \sum_{b=1}^{B} \alpha_b\, d_{j,t-1}^{(b)} [z_{j,t-1} - h_{j,b}] \right) + h_{l,0} + \sum_{j=1}^{M} c_{lj}\, s_{j,t} + \epsilon_{l,t}$$

$$=: \sum_{j=1}^{M} f_{l,j}(z_{j,t-1}) + h_{l,0} + \sum_{j=1}^{M} c_{lj}\, s_{j,t} + \epsilon_{l,t} =: F_l(\boldsymbol{z}_{t-1}), \quad l = 1, 2, \cdots, M. \tag{34}$$

Using this, we can write Eq. 22 in the vector form

$$\boldsymbol{z}_t = \big(F_1(\boldsymbol{z}_{t-1}), F_2(\boldsymbol{z}_{t-1}), \cdots, F_M(\boldsymbol{z}_{t-1})\big)^{\mathsf{T}}. \tag{35}$$

We show that every $F_l$ is continuous and so Eq. 22 is a continuous PWL map. For this purpose, by Eq. 34, it suffices to prove that every $f_{l,j}(z_{j,t-1})$ is continuous. According to the definition of the intervals $I_{j,b}$, Eq. 29, for any $j \in \{1, 2, \cdots, M\}$ we have

$$z_{j,t-1} \in I_{j,1} \quad \Rightarrow \quad d_{j,t-1}^{(b)} = 0 \quad \forall b = 1, 2, \cdots, B,$$

$$z_{j,t-1} \in I_{j,s} \quad \Rightarrow \quad \begin{cases} d_{j,t-1}^{(b)} = 1, & b = 1, 2, \cdots, s-1 \\ d_{j,t-1}^{(b)} = 0, & b = s, s+1, \cdots, B \\ \quad s = 2, 3, \cdots, B, \end{cases}$$

$$z_{j,t-1} \in I_{j,B+1} \quad \Rightarrow \quad d_{j,t-1}^{(b)} = 1 \quad \forall b = 1, 2, \cdots, B. \tag{36}$$

Hence, for $l, j = 1, 2, \cdots, M$, each function $f_{l,j}(z_{j,t-1})$ can be stated as

$$f_{l,j}(z_{j,t-1}) = \begin{cases} f_{l,j}^{(1)} = a_{lj}\, z_{j,t-1}; & z_{j,t-1} \in I_{j,1} \\[4pt] f_{l,j}^{(2)} = (a_{lj} + \alpha_1 w_{lj})\, z_{j,t-1} - \alpha_1 w_{lj} h_{j,1}; & z_{j,t-1} \in I_{j,2} \\[4pt] \vdots \\[4pt] f_{l,j}^{(B)} = (a_{lj} + w_{lj} \sum_{b=1}^{B-1} \alpha_b)\, z_{j,t-1} - w_{lj} \sum_{b=1}^{B-1} \alpha_b h_{j,b}; & z_{j,t-1} \in I_{j,B} \\[4pt] f_{l,j}^{(B+1)} = (a_{lj} + w_{lj} \sum_{i=1}^{B} \alpha_b)\, z_{j,t-1} - w_{lj} \sum_{b=1}^{B} \alpha_b h_{j,b}; & z_{j,t-1} \in I_{j,B+1} \end{cases} \tag{37}$$

Since for every $b = 1, 2, \cdots, B$,

$$\lim_{z_{j,t-1} \to h_{j,b}} f_{l,j}^{(b)}(z_{j,t-1}) = \lim_{z_{j,t-1} \to h_{j,b}} f_{l,j}^{(b+1)}(z_{j,t-1}) = f_{l,j}^{(b)}(h_{j,b}), \tag{38}$$

each function $f_{l,j}(z_{j,t-1})$ is continuous. Hence, Eq. 22 is a continuous PWL map in $\boldsymbol{z}$ (but has discontinuities in its Jacobian matrix across the borders). Because of these properties, all the results established for standard PLRNNs in Monfared & Durstewitz (2020a;b); Schmidt et al. (2021) apply to the dendPLRNN as well, only that the sub-regions and discontinuity boundaries are different. $\square$

### 6.5.5 PROOF OF PROPOSITION 1

*Proof.* Defining $\tilde{z}_t$ as $B$ identical copies of $z_t$,

$$\tilde{z}_t = \begin{pmatrix} \tilde{z}_{1,t} \\ \tilde{z}_{2,t} \\ \vdots \\ \tilde{z}_{M,t} \\ \tilde{z}_{M+1,t} \\ \vdots \\ \tilde{z}_{BM,t} \end{pmatrix} := \begin{pmatrix} z_t \\ z_t \\ \vdots \\ z_t \end{pmatrix}_{BM \times 1} \tag{39}$$

and likewise

$$\tilde{h} = \begin{pmatrix} \tilde{h}_1 \\ \tilde{h}_2 \\ \vdots \\ \tilde{h}_M \\ \tilde{h}_{M+1} \\ \vdots \\ \tilde{h}_{BM} \end{pmatrix} = \begin{pmatrix} h_1 \\ h_2 \\ \vdots \\ h_B \end{pmatrix}_{BM \times 1} , \qquad \tilde{h}_0 = \begin{pmatrix} \tilde{h}_{0,1} \\ \tilde{h}_{0,2} \\ \vdots \\ \tilde{h}_{0,M} \\ \tilde{h}_{0,M+1} \\ \vdots \\ \tilde{h}_{0,BM} \end{pmatrix} = \begin{pmatrix} h_0 \\ h_0 \\ \vdots \\ h_0 \end{pmatrix}_{BM \times 1}$$

$$\tilde{A}_{BM \times BM} = diag\big( \underbrace{A_{M \times M}, A_{M \times M}, \cdots, A_{M \times M}}_{B \text{ times}} \big),$$

$$\tilde{W}_{BM \times BM} = \begin{pmatrix} \alpha_1 W_{M \times M} & \alpha_2 W_{M \times M} & \cdots & \alpha_B W_{M \times M} \\ \hline \alpha_1 W_{M \times M} & \alpha_2 W_{M \times M} & \cdots & \alpha_B W_{M \times M} \\ \vdots & \vdots & \ddots & \vdots \\ \hline \alpha_1 W_{M \times M} & \alpha_2 W_{M \times M} & \cdots & \alpha_B W_{M \times M} \end{pmatrix},$$

$$\tilde{C}s_t = \begin{pmatrix} \tilde{cs}_{1,t} \\ \tilde{cs}_{2,t} \\ \vdots \\ \tilde{cs}_{M,t} \\ \tilde{cs}_{M+1,t} \\ \vdots \\ \tilde{cs}_{BM,t} \end{pmatrix} = \begin{pmatrix} Cs_t \\ Cs_t \\ \vdots \\ Cs_t \end{pmatrix}_{BM \times 1} , \qquad \tilde{\epsilon}_t = \begin{pmatrix} \tilde{\epsilon}_{1,t} \\ \tilde{\epsilon}_{2,t} \\ \vdots \\ \tilde{\epsilon}_{M,t} \\ \tilde{\epsilon}_{M+1,t} \\ \vdots \\ \tilde{\epsilon}_{BM,t} \end{pmatrix} = \begin{pmatrix} \epsilon_t \\ \epsilon_t \\ \vdots \\ \epsilon_t \end{pmatrix}_{BM \times 1} \tag{40}$$

one can rewrite the dendPLRNNfrom Eq. 20 as

$$\tilde{z}_t = \tilde{A}\tilde{z}_{t-1} + \tilde{W} \max(0, \tilde{z}_{t-1} - \tilde{h}) + \tilde{h}_0 + \tilde{C}s_t + \tilde{\epsilon}_t. \tag{41}$$

Now performing the substitution

$$\forall t \qquad \hat{z}_t \leftarrow \tilde{z}_t - \tilde{h}, \tag{42}$$

Eq. 41 can be rewritten as the $M \times B$-dimensional "conventional" PLRNN Eq. 5 with

$$\hat{h}_0 = (\tilde{A} - I)\tilde{h} + \tilde{h}_0. \tag{43}$$

$\square$

### 6.5.6 Proof of Theorem 2

*Proof.* It can easily be shown that for every $i \in \{1, 2, \cdots, M\}$

$$\alpha_b \big[ \max(\max(0, z_{i,t-1} - h_{i,b}) - \max(0, z_{i,t-1})] \in \begin{cases} [-\alpha_b h_{i_b}, 0] \text{ if } \mathrm{sgn}(\alpha_b) = \mathrm{sgn}(h_{i,b}) \\ [0, \alpha_b h_{i,b}] \text{ else} \end{cases} . \tag{44}$$

By defining

$$\sum_{b=1}^{B} \alpha_b \big[ \max(0, \boldsymbol{z}_{t-1} - \boldsymbol{h}_b) - \max(0, \boldsymbol{z}_{t-1})] := \tilde{\phi}(z_{t-1}) = \left( \tilde{\phi}_1(z_{t-1}), \cdots, \tilde{\phi}_M(z_{t-1}) \right)^{\mathsf{T}}, \tag{45}$$

and

$$c_{i,b}^{\mathrm{up}} = \begin{cases} 0 \text{ if } \mathrm{sgn}(\alpha_b) = \mathrm{sgn}(h_{i,b}) \\ \alpha_b h_{i,b} \text{ else} \end{cases} , \qquad c_{i,b}^{\mathrm{low}} = \begin{cases} -\alpha_b h_{i,b} \text{ if } \mathrm{sgn}(\alpha_b) = \mathrm{sgn}(h_{i,b}) \\ 0 \text{ else} \end{cases} , \tag{46}$$

we can conclude that

$$c_i^{low} \le \tilde{\phi}_i(z_{t-1}) \le c_i^{up},$$

where $c_i^{low/up} = \sum_{b=1}^{B} c_{i,b}^{low/up}$. For $c_i = \max\{|c_i^{low}|, |c_i^{up}|\}$ we have

$$\tilde{\phi}_i(z_{t-1})^2 \le c_i^2,$$

and so letting $c = \max\{c_1, c_2, \cdots, c_M\}$ yields

$$\left\| \tilde{\phi}(z_{t-1}) \right\| = \sqrt{\sum_{i=1}^{M} \left( \tilde{\phi}_i(z_{t-1}) \right)^2} \le \sqrt{\sum_{i=1}^{M} c^2} := \tilde{c}. \tag{47}$$

Since

$$\boldsymbol{z}_t = \boldsymbol{A}\,\boldsymbol{z}_{t-1} + \boldsymbol{W}\,\tilde{\phi}(\boldsymbol{z}_{t-1}) + \boldsymbol{h}_0, \tag{48}$$

for $T \in \mathbb{N}$ and $t = 2, \cdots, T$, computing $z_2, z_3, \cdots, z_T$ recursively leads to

$$\boldsymbol{z}_2 = \boldsymbol{A}\,\boldsymbol{z}_1 + \boldsymbol{W}\,\tilde{\phi}(\boldsymbol{z}_1) + \boldsymbol{h}_0$$

$$\boldsymbol{z}_3 = \boldsymbol{A}^2\,\boldsymbol{z}_1 + \boldsymbol{A}\,\boldsymbol{W}\,\tilde{\phi}(\boldsymbol{z}_1) + \boldsymbol{W}\,\tilde{\phi}(\boldsymbol{z}_2) + \big[\boldsymbol{A} + \boldsymbol{I}\big]\boldsymbol{h}_0$$

$$\vdots$$

$$\boldsymbol{z}_T = \boldsymbol{A}^{T-1}\,\boldsymbol{z}_1 + \sum_{j=0}^{T-2} \boldsymbol{A}^j\,\boldsymbol{W}\,\tilde{\phi}(\boldsymbol{z}_{T-1-j}) + \sum_{j=0}^{T-2} \boldsymbol{A}^j\,\boldsymbol{h}_0. \tag{49}$$

Therefore, by Eq. 47, for every $T \ge 2$, we have

$$\|\boldsymbol{z}_T\| \le \|\boldsymbol{A}\|^{T-1}\,\|\boldsymbol{z}_1\| + \tilde{c}\,\|\boldsymbol{W}\| \sum_{j=0}^{T-2} \|\boldsymbol{A}\|^j + \sum_{j=0}^{T-2} \|\boldsymbol{A}\|^j\,\|\boldsymbol{h}_0\|. \tag{50}$$

If $\sigma_{\max}(\boldsymbol{A}) < 1$, then $\lim_{T \to \infty} \|\boldsymbol{A}\|^{T-1} = 0$ and

$$\lim_{T \to \infty} \|\boldsymbol{z}_T\| \le \tilde{c}\,\|\boldsymbol{W}\| \sum_{j=0}^{\infty} \|\boldsymbol{A}\|^j + \sum_{j=0}^{\infty} \|\boldsymbol{A}\|^j\,\|\boldsymbol{h}_0\| = \frac{\tilde{c}\,\|\boldsymbol{W}\| + \|\boldsymbol{h}_0\|}{1 - \|\boldsymbol{A}\|} < \infty. \tag{51}$$

$\square$

