# OpenReview forum: "Tractable Dendritic RNNs for Identifying Unknown Nonlinear Dynamical Systems"
_ICLR.cc/2022/Conference — ICLR 2022 Submitted_

### Official Review · Reviewer_caaB · 2021-10-30

**Correctness:** 3
**Technical Novelty And Significance:** 2
**Empirical Novelty And Significance:** 1
**Recommendation:** 3
**Confidence:** 4

**Main Review:**

I think the paper is well-written and easy to follow. The spline basis extension is straightforward and explained well by comparing it with PLRNN. However, I think the overall novelty is not that strong enough. The spline basis expansion is quite common in many problems, like in mixture models, instead of using one model, people extend to mixture models with a linear combination; in kernel study, instead of using a single kernel, people use a mixture of kernels. In early papers such as [Chan et al ACC 1998], people have studied spline bases to model the nonlinear relationship between input and outputs in RNN. Splines are a common choice of basis functions to capture nonlinear functional relationships in a variety of neural systems [Qian et al Neural Computation 2018, Huang et al Neural Computation 2009, Frank et al Journal of Neuro 2002]. Therefore, although spline basis performs well in the context of PLRNN, I don't think the technical contribution is much significant to the community of ICLR.

Some detailed comments:

1. It's claimed that "we can essentially reformulate a high-dimensional PLRNN in terms of an equivalent lower-dimensional dendPLRNN". I think the equivalence is only true when the high-dimensional PLRNN has a unique structure of the connection weight W as shown in eq. 40. Therefore, although a dendPLRNN can be formulated as a high-dimensional PLRNN, the opposite direction is not always true.

2. Following point 1, at the end of page 6, "it becomes clear that the basis expansion enables to reduce the model’s overall dimensionality without compromising performance." I think this is only true if the structured assumption of W is valid. If the authors want to compare PLRNN and dendPLRNN in the same space, e.g., M*B, it's needed to show that how much approximation error there will be.















**Summary Of The Paper:**

The authors proposed an extension of PLRNN with a dendrite-like formulation of the nonlinear activation function. They showed that the proposed dendPLRNN maintains the mathematical tractability and dynamical systems interpretation as PLRNN. They compared with alternatives for DS reconstruction with an overall better performance.


**Summary Of The Review:**

The spline basis expansion works empirically but the technical novelty is not strong enough.

---

> ### Author Response · Authors · 2021-11-23
> **Point by point reply**
>
> The “lack of novelty argument” is notoriously hard to deal with, of course (it gives little guidance on how to move on with a manuscript). We are amply aware that similar strategies have been applied elsewhere. In fact, we had referred to its common use in statistics in sect. 3.2 already, and have now further expanded on this based on the referee’s citations (except Chan et al. 98, which we could not identify). But we also would like to remark that we feel that this literature is only peripherally related to our work here, as it primarily deals with statistical and data-analytical applications in neuroscience.
>
> This specific application for the purpose of DS reconstruction, however, we think is still very novel. And it’s not a mere addition of a known technique to an existing algorithm, but it is in fact a carefully chosen one which makes the model more expressive while not compromising its tractability, worked out with a new set of propositions and theorems, further mathematical results in Appx. 6.5, and empirical validation in sect. 4.3. We also would like to highlight that in this context we have collected a number of benchmark systems with different properties and a set of performance evaluation measures, a database for benchmarking that has not been around in this form before, to which we have added another real-world example now, and that will facilitate future research in this area in our minds.
>
> *Response to detailed comments*
>
> 1. It is true that formally we have shown only the direction “dendPLRNN → conventional PLRNN”. Vice versa, the conventional PLRNN is of course just a special case of a dendPLRNN, but as the referee rightly points out, not necessarily a lower-dimensional one. But we conjecture that this will indeed be the case much more generally, beyond the specific form used in the proof of Theorem 2 in Appx. 6.5.5 (to formally show this, of course, one would need to find parameters for a lower-dim. dendPLRNN such that the outputs of its dynamical variables exactly equal those of a properly defined subsystem of the regular PLRNN for each time step). For now we have rephrased the respective parts more cautiously, and rely on our empirical validation of this point in sect. 4.3.
>
> 2. We are not sure we got this right (we think this is what we actually have empirically demonstrated in sect. 4.3), but will try to respond based on our understanding. With dimensionality we are referring to the size of the latent space, $M$, the number of dynamical variables. If the dynamics can be represented in a lower-dimensional state space, this will profoundly ease any dynamical systems analysis (in fact, many theoretical concepts in dynamical systems theory have been introduced just for this purpose, like e.g. Poincare sections). That this is the case, we had confirmed empirically.
> At the same, we empirically demonstrated that the dendPLRNN also gets away with an usually much lower number of parameters *at the same reconstruction quality* (as assessed by our various measures). So we think what the referee suggests here is something we had actually done, no?

---

### Official Review · Reviewer_VV4w · 2021-11-02

**Correctness:** 4
**Technical Novelty And Significance:** 4
**Empirical Novelty And Significance:** 4
**Recommendation:** 6
**Confidence:** 4

**Main Review:**

Post-rebuttal: Thanks for addressing my concerns. The authors have updated the manuscript and added experiments. I will not change my recommendation.

#####
Pros:

+ Capacity to capture nonlinear dynamics

+ Competitive performance

+ Simple form and better interpretability

Concerns:

* As a modification to PLRNN, PLRNN could be a baseline and comparing to it would be confirmative.

* SINDy and LSTM require fully observed data. Do the authors consider to compare with other SSM methods like LFADS?

* To demonstrate the expressive power, apart from chaotic attractors, there are many dynamical systems such as fixed attractors, continuous attractors and etc that are commonly used to describe e.g. neural computational (Wang 2006, Mante 2013). Such experiments could be useful for qualitative demonstration.

* Lack for real-world data example.

**Summary Of The Paper:**

The paper proposes to modify the nonlinearity of neuron couplings in PLRNN with a linear spline basis expansion. This modification is claimed to boost PLRNN's capacity of capturing arbitrary nonlinear dynamics in low-dimensions. The paper also introduces two training frameworks, variational inference and BPTT. The proposed method is evaluated on various nonlinear dynamical systems.

**Summary Of The Review:**

Overall, I vote for accepting. The paper is well written. The method is evaluated on various data and compared to other methods. Hopefully the authors can address my concern in the rebuttal period.

---

> ### Author Response · Authors · 2021-11-23
> **Point by point reply to concerns**
>
> 1) A plain PLRNN has now been included for comparison as suggested (new Table 2).
>
>
> 2) We have considered comparing with LFADS, but in its original design LFADS focuses on posterior inference, as the data is incorporated into the generative model via the controller variable at each time step. LFADS could be turned into a fully generative model of the underlying dynamics by removing the controller, but the published code is only implemented in an older Python+Tensorflow version, which we did not manage to modify and get running in the period available for the revision.
>
>
> 3) This is true, there are other interesting and neuroscientifically relevant scenarios, some of them we had addressed in past publications. Here, however, to push the methods (and avoid a ‘ceiling effect’), we intentionally chose to pick very challenging systems, that are chaotic, high-dimensional and noisy, or contain very different time scales (bursting neuron model).
> Instead, we have added now an empirical example which is also very challenging.
>
>
> 4) Essentially the whole theoretical literature on nonlinear dynamical systems reconstruction focuses on simulated benchmark models, as their properties are known and easy to control.
> Nevertheless, as suggested, we now have added a real-world dataset (EEG recordings), with the reconstruction performance illustrated in new Fig. S5, and the systematic comparison between all methods on this dataset added to Table 1.

---

### Official Review · Reviewer_B5hP · 2021-11-08

**Correctness:** 2
**Technical Novelty And Significance:** 2
**Empirical Novelty And Significance:** 2
**Recommendation:** 5
**Confidence:** 4

**Main Review:**

The motivation of the model design in this paper is ambiguous. The proposed model is a simplified hybrid linear and nonlinear dynamic system. The reasoning of coupling this model with dendritic computation is not clear. The claimed interpretability of the proposed model lacks both theoretical and experimental justification. The tractability of the training and inference method seems to stem from the VI and BPTT, thereby making the paper lack technical contribution.

Eq. (2) and (4) refer to the same function denoted by \phi, which incurs confusing definitions. Which function is exactly used in the model?

The proposed dynamical system is termed with piecewise-linear RNN. In time series analysis, "piecewise" often refers to the method that separately models the segments of time series [1]. In this paper, the implication of "piecewise" is not well explained.

One of the training methods applied in this paper is based on VI. Since VI on time series/sequence is heavily studied in recent research works, e.g. [2-5], the authors are expected to detail the design of their VI based training and the comparison to existing methods.

In the experiment section, the result is unconvincing in that the proposed method failed to exhibit consistent out-performance on the metrics/datasets, i.e. mostly only on half of the datasets, the proposed methods can outperform.

[1] 2007, SDM. A better alternative to piecewise linear time series segmentation

[2] 2015, NeurIPS, A Recurrent Latent Variable Model for Sequential Data

[3] 2016, NIPS, Sequential Neural Models with Stochastic Layers

[4] 2017, AAAI, Structured Inference Networks for Nonlinear State Space Models

[5] 2021, ICLR, Mind the Gap when Conditioning Amortised Inference in Sequential Latent-Variable Models


**Summary Of The Paper:**

This paper focuses on the nonlinear dynamical system on time series data. It proposed a piecewise-linear (PL) recurrent neural network (RNN) and attempted to couple it with dendritic computation, a computing paradigm from the neural computation discipline. The training method of the proposed model is through variational inference, and BPTT with teacher forcing.


**Summary Of The Review:**

The motivation of this paper needs more clarification. The proposed model is a simple hybrid dynamic system and the training method is based on existing VI and BPTT and thus the paper lacks novelty and technical contribution. The experiment is not solid enough to support the claim advantages of the model, i.e. interpretability and approximating ability.

---

> ### Author Response · Authors · 2021-11-23
> **reply to points 1-4**
>
> *1) Motivation and tractability not clear*
>
> There are several fundamental misunderstandings here: The training algorithms (BPTT and VI) are completely secondary. Of course these have been around for a while and are not our contributions, and they are not related to the model’s tractability/ interpretability in any way.
>
> The tractability comes from the model’s (dend-PLRNN) mathematical form. In dynamical systems reconstruction we are aiming for a generative model of the dynamical system that underlies the observed time series. Ideally, after training, we would like to be able to easily analyze these models for their dynamical properties and mechanisms. This is hugely important in science since ultimately we would like to understand what are the mechanisms behind the observed time series. With most other approaches for dynamical systems reconstruction this is not easily possible, however: They are mostly black box and require reverse engineering after training (e.g. Barak & Sussillo 2013).
>
> For the dend-PLRNN this is different: As summarized in sect. 3.3, many of its dynamical properties like fixed points, cycles, and their stability, can be computed analytically, and are hence easily accessible. This is what we mean by tractability, and we have worked this out explicitly mathematically in sect. 6.4. We also had used this mathematical tractability in the experimental section, Fig. 2, for instance, where analytically computed fixed points were provided.
>
> Thus, interpretability was indeed demonstrated both theoretically and experimentally, and the proposed innovations are not the training algorithms of course, but the mathematical model setup which combines a highly expressive formulation with easy mathematical tractability. This was indeed the major motivation for this work.
>
> *2) Eq. 2 vs. 4*
>
> As stated in the text, eq. (2) was replaced by eq. (4) for our model. Note that eq. (2) is a special case of eq. (4) of course. We acknowledge that this could potentially be confused, but on the other hand find it important to highlight the general structure of model eq. (1) (independent of the specific nonlinearity plugged in for $\phi$), to which we would like to be able to refer back to in later sections. We hope this makes sense to the referee.
>
> *3) Piecewise-linear*
>
> The term piecewise-linear RNN (PLRNN) has been established previously in the literature (see for instance Schmidt et al. 2021, https://openreview.net/forum?id=_XYzwxPIQu6). Formally, it is a piecewise-linear (PWL) map due to the ReLU activation function, which makes this a system of piecewise linear functions in a strict and well-defined mathematical sense (we just used the common mathematical definition here, nothing else). The definition in time series analysis/statistics generally employs this same definition, e.g. for splines (as we had pointed out in sect. 3.3).
>
> *4) Variational Inference*
>
> Our paper deals with nonlinear dynamical systems reconstruction, not with VI. VI is merely used as one of our training algorithms. The proper set of comparison methods therefore comes from the area of nonlinear dynamical systems reconstruction, of which we provided several. Note that not any VI-based algorithm for sequential data is also automatically suited for DS reconstruction, and in fact this was neither the goal nor has this been demonstrated in any of the papers cited.
>
> Please also note that the details of our VI design were given in Appx. A.6.1, and that ref. [2] and [4] were indeed already cited in our original submission. We have added the other suggested references as well, but emphasize that in the present context which deals with dynamical systems reconstruction the VI literature is not the most relevant one in our minds, but more the lit. on DS reconstruction summarized in sect. 2.

---

> > ### Author Response · Authors · 2021-11-23
> > **Reply to point 5 and summary evaluation**
> >
> > *5) Results unconvincing*
> >
> > Our method mostly outperforms the other methods, except for SINDy. But note that Table 1 is only part of the story, and the results need to be interpreted in context: As explained in sect. 4.4, unlike our method, SINDy (although it's certainly an appealing and fast method) can essentially only handle dynamical systems for which the general functional form (e.g., polynomial) *is known a priori*, since the library must contain the right terms for SINDy to learn properly in many cases. For that reason, SINDy completely failed on the bursting neuron and neural population benchmarks. Our method does not have this restriction: It can learn any type of underlying dynamical system without any prior knowledge or assumptions about its functional form, which is a huge advantage!
> >
> > Moreover, our system is the *only one* which is mathematically tractable in the dynamical systems sense, as explained above, in sect. 3.3, and worked out in sect. 6.4. This again is a profound advantage over the other methods in any scientific setting where we seek to understand the dynamical mechanisms underlying the time series. These are additional and important advantages that one needs to consider in evaluating the different models, beyond the mere performance comparisons provided by Table 1.
> > Finally, we have now added a comparison of all models on a real-world dataset (EEG recordings), on which it also outperforms all other methods (see updated Table 1), incl. SINDy which - once again - is not capable to learn the underlying dynamics.
> >
> >
> > *Summary*
> >
> > See replies above: The motivation for this novel model architecture was the combination of mathematical tractability with high model capacity (this is also what makes this novel, together with the whole of sect. 3.3; this is not a paper on either VI or BPTT). The powerful approximation capabilities of our model are illustrated in Figs. 2 (how could one even improve on that?) and 3, and it’s analytical tractability was proven in Theorems 1 & 2, Proposition 1, and the whole of sect. 6.4.
> > We also would like to remark that we do not quite understand the low correctness score the referee has assigned to our work, since s/he did not point out any major flaws in our methodology, math. proofs or in the technical parts.

---

### Author Response · Authors · 2021-11-23
**General reply to all referees**

We thank all referees for their feedback which we address in detail below in our point-to-point replies.

We have added the following new results to our manuscript:
- We added a comparison between our dendPLRNN and the plain (standard) PLRNN trained both with VI and TF (Table 2 in Appx. 6.3).
- We included a model comparison on real-world data, namely EEG (electrophysiological) time series (updated Table 1, new Fig. S5).
- We added the suggested literature.
- We modified the text to incorporate these new results, and clarify some of the misunderstandings.

---

### Decision · Program_Chairs · 2022-01-20

**Decision:**

Reject

**Comment:**

Inspired by dendritic nonlinearity, this paper extends previous work on PLRNN/PWL dynamical system modeling by Durstewitz's group. The extension replaces the ReLU nonlinearity with a linear combination of ReLUs. This preserves the theoretical properties of PLRNN, however, the dimensionality of the latent dynamics remains the same, increasing the expressive power of prior PLRNNs. I (area chair) actually read this paper since not all reviewers provided high-quality reviews and one key reviewer is having a personal emergency. Though I appreciate the premise, detailed numerical evaluations, and the inference approach, the novelty is marginal and I do not buy the theoretical advantage of this class of models as presented (see below). Therefore I cannot recommend this paper to appear at ICLR at this time.

Some additional weaknesses that reviewers did not point out:
1. Dendritic nonlinearity is summarized as a point nonlinearity; It lacks the interesting phenomena of dendrites such as nonlinear summation and calcium spikes with its own internal dynamics.
2. The many analytical properties of PLRNN may sound nice on paper, but very impractical. To search for the fixed points and cycles, the amount of required computation exponentially increases as the number of neurons and cycle length increases. In addition the boundary effects cannot always be ignored. In general detailed analysis can become quite non-trivial quickly, e.g., https://arxiv.org/abs/2109.03198
3. High-dimensional PLRNN that approximates a low-dimensional dynamical system due to model mismatch won't have the same topological stability structures. Theoretical analysis of higher-dimensional DS may be very misleading.

---

> ### Public Comment · ~Daniel_Durstewitz1 · 2022-02-05
> **Public reply to additional points raised in decision note**
>
> We thank the AC for the reading and additional comments on our manuscript (and also for acknowledging that some of the reviews we received weren’t of particularly high quality). We surely appreciate the feedback. However, if we understood correctly, these additional points apparently influenced the AC’s final decision on our paper. In this case it would have been good if we would have had the opportunity to respond to them during the actual review process.
>
> So, at the very least, we would like to do this now – in brief:
>
> 1. We implemented one hallmark feature of dendritic processing which is the combination of several nonlinear processing subunits at the soma. We completely agree with the AC that dendrites have many other interesting biophysical and dynamical properties that might be very worth mimicking, but – on the other hand – clearly this was not the goal of the present study. In our study we did not aim for a biologically detailed model of dendrites, but rather only wanted to exploit one core feature in our context of DS reconstruction.
>
> 2. What are the alternatives? Much effort has been invested on how to (approximately, numerically) locate fixed points of RNNs during (e.g. Duncker et al. 2019, Smith et al. 2021) or after (e.g. Sussillo & Barak 2013) training. Although the AC is correct that in principle we are dealing with a combinatorial problem here (which can become an issue in very high dimensions), in practice for all PLRNNs we have inferred from benchmark or real-world data so far, we were able to *exactly* locate *all* fixed points (and assess their stability) in a matter of (split-)seconds on a single CPU, and we have used this property widely both in previous (e.g. Schmidt et al. 2021, ICLR) and the present (Fig. 2) paper. We do think this alone is a huge advantage over other approaches! Cycles are a bit more tricky (and usually only assessed numerically, if stable), while for the PLRNN there are efficient numerical algorithms (similar in spirit to the Simplex algo) which with probability approaching 1 will find all cycles up to quite high order (whether stable or not). There are many other advantages (some of them alluded to in our theoretical section). In general, there is simply a large body of work on PWL maps, and there are good reasons why PWL models have been so popular in engineering and science over decades.
>
> 3. This is a mere “conjecture”, not a fact (it is also one that would equally apply to essentially any other current DS reconstruction approach, not just ours!). Note that the recovered (and true) DS will usually occupy a much lower-dimensional manifold in the system’s state space, such that the nominal dimensionality is of less importance. In fact, this is exactly the case for the embedding and delay reconstruction theorems (Whitney; Takens; Sauer et al.): embeddings/ delay reconstructions most commonly will *not* have the same nominal dimensionality as the original system, yet topological equivalence (diffeomorphism) is guaranteed by the theorems. Also, while in general we agree that more theoretical work is needed to establish the correspondence of true and reconstructed systems (for any such approach, not just ours!), just empirically, we found that reconstructed invariant sets like the Lorenz attractor mostly retained their topological and geometrical properties (including stability) in the reconstructed state spaces (e.g., as assessed by our Kullback-Leibler measure).